# Grounding Aleatoric Uncertainty for Unsupervised Environment Design

**Minqi Jiang**[*]
UCL & Meta AI

**Michael Dennis**
UC Berkeley

**Jack Parker-Holder**
University of Oxford

**Andrei Lupu**
MILA & Meta AI

**Heinrich Küttler**[‡]
Inflection AI

**Edward Grefenstette**[‡]
UCL & Cohere

**Tim Rocktäschel**[‡]
UCL

**Jakob Foerster**
FLAIR, U of Oxford

## Abstract

Adaptive curricula in reinforcement learning (RL) have proven effective for producing policies robust to discrepancies between the train and test environment. Recently, the Unsupervised Environment Design (UED) framework generalized RL curricula to generating sequences of entire environments, leading to new methods with robust minimax regret properties. Problematically, in partially-observable or stochastic settings, optimal policies may depend on the ground-truth distribution over aleatoric parameters of the environment in the intended deployment setting, while curriculum learning necessarily shifts the training distribution. We formalize this phenomenon as *curriculum-induced covariate shift* (CICS), and describe how its occurrence in aleatoric parameters can lead to suboptimal policies. Directly sampling these parameters from the ground-truth distribution avoids the issue, but thwarts curriculum learning. We propose SAMPLR, a minimax regret UED method that optimizes the ground-truth utility function, even when the underlying training data is biased due to CICS. We prove, and validate on challenging domains, that our approach preserves optimality under the ground-truth distribution, while promoting robustness across the full range of environment settings.

## 1 Introduction

Adaptive curricula, which dynamically adjust the distribution of training environments to facilitate learning [26, 35], have played a key role in many recent achievements in deep reinforcement learning (RL). Applications span both single-agent RL [34, 53, 58, 22, 27], where adaptation occurs over environment variations, and multi-agent RL, where adaptation can additionally occur over co-players [43, 52, 44]. These methods demonstrably improve the sample efficiency and robustness of the final policy [25, 8, 21, 20], e.g. by presenting the agent with challenges at the threshold of its abilities.

In this paper we introduce and address a fundamental problem relevant to adaptive curriculum learning methods for RL, which we call *curriculum-induced covariate shift* (CICS). Analogous to the covariate shift that occurs in supervised learning (SL) [18], CICS refers to a mismatch between the *input distribution* at training and test time. In the case of RL, we will show this becomes problematic when the shift occurs over the *aleatoric parameters* of the environment—those aspects of the environment holding irreducible uncertainty even in the limit of infinite experiential data [9]. While in some cases, CICS may impact model performance in SL, adaptive curricula for SL have generally not been found to be as impactful as in RL [55]. Therefore, we focus on addressing CICS specifically as it arises in the RL setting, leaving investigation of its potential impact in SL to future work.

---

[*]Correspondence to msj@meta.com. ‡ Work done while at Meta AI.

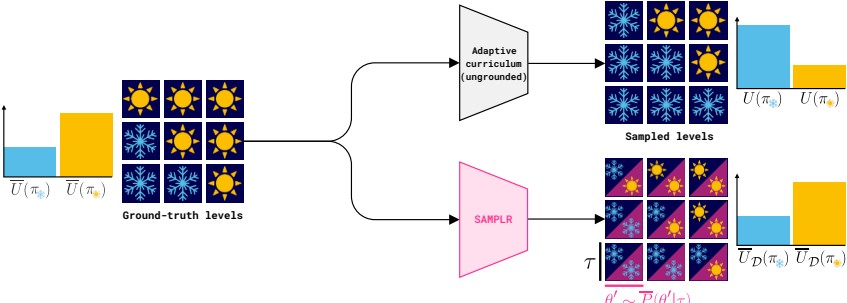

Figure 1: Adaptive curricula can result in covariate shifts in environment parameters with respect to the ground-truth distribution $\overline{P}(\Theta)$ (top path), e.g. whether a road is icy or not, which can cause the policy to be optimized for a utility function $U$ differing from the ground-truth utility function $\overline{U}$ based on $\overline{P}$ (See Equation 1). Here, the policies $\pi_{\text{❄}}$ and $\pi_{\text{☀}}$ drive assuming ice and no ice respectively. SAMPLR (bottom path) matches the distribution of training transitions to that under $\overline{P}(\Theta|\tau)$ (pink triangles), thereby ensuring the optimal policy trained under a biased curriculum retains optimality for the ground-truth distribution $\overline{P}$.

To establish precise language around adaptive curricula, we cast our discussion under the lens of Unsupervised Environment Design [UED, 8]. UED provides a formal problem description for which an optimal curriculum is the solution, by defining the *Underspecified POMDP* (UPOMDP; see Section 2), which expands the classic POMDP with a set of *free parameters* $\Theta$, representing the aspects of the environment that may vary. UED then seeks to adapt distributions over $\Theta$ to maximize some objective, potentially tied to the agent's performance. UED allows us to view adaptive curricula as emerging via a multi-player game between a *teacher* that proposes environments with parameters $\theta \sim P(\Theta)$ and a *student* that learns to solve them. In addition to notational clarity, this formalism enables using game theoretic constructs, such as Nash equilibria [NE, 28], to analyze curricula.

This game-theoretic view has led to the development of curriculum methods with principled robustness guarantees, such as PAIRED [8] and Robust Prioritized Level Replay [PLR$^{\perp}$, 20], which aim to maximize a student's regret and lead to minimax regret [40] policies at NE. Thus, at NE, the student can solve all solvable environments within the training domain. However, in their current form the UED robustness guarantees are misleading: if the UED curriculum deviates from a ground-truth distribution $\overline{P}(\Theta)$ of interest, i.e. the distribution at deployment, with respect to aleatoric parameters $\Theta' \subset \Theta$, the resulting policies may be suboptimal under the ground-truth distribution $\overline{P}$.

For a concrete example of how CICS can be problematic, consider the case of training a self-driving car to navigate potentially icy roads, when icy conditions rarely occur under $\overline{P}$. When present, the ice is typically hard to spot in advance; thus, the aleatoric parameters $\Theta'$ correspond to whether each section of the road is icy. A priori, a curriculum should selectively sample more challenging icy settings to facilitate the agent's mastery over such conditions. However, this approach risks producing an overly-pessimistic agent (i.e. one that assumes that ice is common), driving slowly even in fair weather. Such a policy leads to inadequate performance on $\overline{P}$, which features ice only rarely.

We can preserve optimality on $\overline{P}$ by *grounding the policy*—that is, ensuring that the agent acts optimally with respect to the *ground-truth utility function* for any action-observation history $\tau$ and the implied ground-truth posterior over $\Theta$:

$$\overline{U}(\pi|\tau) = \mathbb{E}_{\theta \sim \overline{P}(\theta|\tau)} \left[ \overline{U}(\pi|\tau, \theta) \right], \tag{1}$$

where the ground-truth utility conditioned on $X$, $\overline{U}(\pi|X)$, is defined to be $\mathbb{E}_{\tau, \theta \sim \overline{P}(\theta|X)} [\sum_{t=0}^{\infty} \gamma^t r_t]$, for rewards $r_t$ and a discount $\gamma$.

We can ground the policy by *grounding the training distribution*, which means constraining the training distribution of aleatoric parameters $P(\Theta')$ to match $\overline{P}(\Theta')$. This is trivially accomplished by directly sampling $\theta' \sim \overline{P}(\Theta')$, which we call *naive grounding*. Unfortunately, this approach makes many curricula infeasible by removing the ability to selectively sample environment settings over aleatoric parameters. Applying this strategy to the self-driving agent may result in a policy that is optimal in expectation under $\overline{P}$ where there is rarely ice, but nevertheless fails to drive safely on ice.

We wish to maintain the ability to bias a training distribution, since it is required for curriculum learning, while ensuring the resulting decisions remain optimal in expectation under $\overline{P}$. This goal is captured by the following objective:

$$\overline{U}_{\mathcal{D}}(\pi) = \mathbb{E}_{\tau \sim \mathcal{D}} \left[ \, \overline{U}(\pi|\tau) \right], \tag{2}$$

where $\mathcal{D}$ is the training distribution of $\tau$. Under naive grounding, $\mathcal{D}$ is equal to $\overline{P}(\tau)$ and Equation 2 reduces to $\overline{U}(\pi)$. To overcome the limitations of naive grounding, we develop an approach that allows $\mathcal{D}$ to deviate from $\overline{P}(\tau)$, e.g. by prioritizing levels most useful for learning, but still grounds the policy by evaluating decisions following potentially biased training trajectories $\tau$ according to $\overline{U}(\pi|\tau)$. Figure 1 summarizes this approach, and contrasts it with an ungrounded adaptive curriculum.

In summary this work presents the following contributions: i) We first formalize the problem of CICS in RL in Section 3. ii) Then, we present SAMPLR, which extends PLR$^{\perp}$, a state-of-the-art UED method, to preserve optimality on $\overline{P}$ while training under a usefully biased training distribution in Section 4. iii) We prove in Section 5 that SAMPLR promotes Bayes-optimal policies that are robust over all environment settings $\theta \sim \overline{P}(\Theta)$. iv) Our experiments validate these conclusions in two challenging domains, where SAMPLR learns highly robust policies, while PLR$^{\perp}$ fails due to CICS.

## 2 Background

### 2.1 Unsupervised Environment Design

Unsupervised Environment Design [UED, 8] is the problem of automatically generating an adaptive distribution of environments which will lead to policies that successfully transfer within a target domain. The domain of possible environment settings is represented by an Underspecified POMDP (UPOMDP), which models each environment instantiation, or *level*, as a specific setting of the *free parameters* that control how the environment varies across instances. Examples of free parameters are the position of walls in a maze or friction coefficients in a physics-based task. Formally a UPOMDP is defined as a tuple $\mathcal{M} = \langle A, O, \Theta, \mathcal{S}, \mathcal{T}, \mathcal{I}, \mathcal{R}, \gamma \rangle$, where $A$ is the action space, $O$ is the observation space, $\Theta$ is the set of free parameters, $S$ is the state space, $\mathcal{T} : S \times A \times \Theta \rightarrow \mathbf{\Delta}(S)$ is the transition function, $\mathcal{I} : S \rightarrow O$ is the observation function, $\mathcal{R} : S \rightarrow \mathbb{R}$ is the reward function, and $\gamma$ is the discount factor. UED typically approaches the curriculum design problem as training a *teacher* agent that co-evolves an adversarial curriculum for a *student* agent, e.g. by maximizing the student's regret.

### 2.2 Prioritized Level Replay

We focus on a recent UED algorithm called Robust Prioritized Level Replay [PLR$^{\perp}$, 21], which performs environment design via random search. PLR$^{\perp}$ maintains a buffer of the most useful levels for training, according to an estimate of learning potential—typically based on regret, approximated by a function of the temporal-difference (TD) errors incurred on each level. For each episode, with probability $p$, PLR$^{\perp}$ actively samples the next training level from this buffer, and otherwise evaluates regret on a new level $\theta \sim \overline{P}(\Theta)$ without training. This sampling mechanism provably leads to a minimax regret policy for the student at NE, and has been shown to improve sample-efficiency and generalization. The resultant regret-maximizing curricula naturally avoid unsolvable levels, which have no regret. We provide implementation details for PLR$^{\perp}$ in Appendix A.

## 3 Curriculum-Induced Covariate Shift

Since UED algorithms formulate curriculum learning as a multi-agent game between a teacher and a student agent, we can formalize when CICS becomes problematic by considering the equilibrium point of this game: Let $\Theta$ be the environment parameters controlled by UED, $\overline{P}(\Theta)$, their ground-truth distribution, and $P(\Theta)$, their curriculum distribution at equilibrium. We use $\tau_t$ to refer to the joint action-observation history (AOH) of the student until time $t$ (and simply $\tau$ when clear from context). Letting $V(\pi|\tau_t)$ denote the value function under the curriculum distribution $P(\Theta)$, we characterize an instance of CICS over $\Theta$ as *problematic* if the optimal policy under $P(\Theta)$ differs from that under the ground-truth $\overline{P}(\Theta)$ for some $\tau_t$, so that

$$\arg\max_{\pi} V(\pi|\tau_t) \neq \arg\max_{\pi} \overline{V}(\pi|\tau_t).$$

The value function $\overline{V}(\pi|\tau_t)$ with respect to $\overline{P}(\Theta)$ can be expressed as a marginalization over $\theta$:

$$\overline{V}(\pi|\tau_t) = \sum_\theta \overline{P}(\theta|\tau_t)\tilde{V}(\pi|\tau_t,\theta) \propto \sum_\theta \overline{P}(\theta)\tilde{P}(\tau_t|\theta)\tilde{V}(\pi|\tau_t,\theta). \qquad (3)$$

Here, the notation $\overline{P}(\theta)$ means $\overline{P}(\Theta = \theta)$, and the tilde on the $\tilde{P}$ and $\tilde{V}$ terms indicates independence from any distribution over $\Theta$, as they both condition on $\theta$. Importantly, the value function under the curriculum distribution $V(\pi|\tau_t)$ corresponds to Equation 3 with $\overline{P}$ replaced by $P$. We see that $\overline{V}(\pi|\tau_t)$ is unchanged for a given $\tau_t$ when $\overline{P}(\theta)$ is replaced with $P(\theta)$ if 1) $\overline{P}(\theta^*|\tau_t) = 1$ for some $\theta^*$, and 2) $\overline{P}$ shares support with $P$. Then $\tilde{P}(\tau_t|\theta) = 1$ iff $\theta = \theta^*$ and zero elsewhere. In this case, the sums reduce to $\overline{V}(\pi|\tau_t,\theta^*)$, regardless of changing the ground-truth distribution $\overline{P}$ to $P$. In other words, when $\Theta$ is fully determined given the current history $\tau$, covariate shifts over $\Theta$ with respect to $\overline{P}(\Theta)$ have no impact on policy evaluation and thus the value function for the optimal policy. If the first condition does not hold, the uncertainty over the value of some subset $\Theta' \subset \Theta$ is irreducible given $\tau$, making $\Theta'$ aleatoric parameters for the history $\tau$. Thus, assuming the curriculum shares support with the ground-truth distribution, covariate shifts only alter the optimal policy at $\tau$ when they occur over aleatoric parameters given $\tau$. Such parameters can arise when the environment is inherently stochastic or when the cost of reducing uncertainty is high.

Crucially, our analysis assumes $P$ and $\overline{P}$ share support over $\Theta$. When this assumption is broken, the policy trained under the curriculum can be suboptimal for environment settings $\theta$, for which $P(\theta) = 0$ and $\overline{P}(\theta) > 0$. In this paper, we specifically assume that $P$ and $\overline{P}$ share support and focus on addressing suboptimality under the ground-truth $\overline{P}$ due to CICS over the aleatoric parameters $\Theta'$.

This discussion thus makes clear that problematic CICS can be resolved by *grounding the training distribution*, i.e. enforcing the constraint $P(\Theta'|\tau) = \overline{P}(\Theta'|\tau)$ for the aleatoric parameters of the environment. This constraint results in *grounding the policy*, i.e. ensuring it is optimal with respect to the ground-truth utility function based on $\overline{P}$ (Equation 1). As discussed, naive grounding satisfies this constraint by directly sampling $\theta' \sim \overline{P}(\Theta')$, at the cost of curricula over $\Theta'$. This work develops an alternative for satisfying this constraint while admitting curricula over $\Theta'$.

## 4 Sample-Matched PLR (SAMPLR)

We now describe a general strategy for addressing CICS, and apply it to PLR$^\perp$, resulting in Sample-Matched PLR (SAMPLR). This new UED method features the robustness properties of PLR$^\perp$ while mitigating the potentially harmful effects of CICS over the aleatoric parameters $\Theta'$.

As discussed in Section 3, CICS become problematic when the covariate shift occurs over some aleatoric subset $\Theta'$ of the environment parameters $\Theta$, such that the expectation over $\Theta'$ influences the optimal policy. Adaptive curriculum methods like PLR$^\perp$ prioritize sampling of environment settings where the agent experiences the most learning. While such a curriculum lets the agent focus on correcting its largest errors, the curriculum typically changes the distribution over aleatoric parameters $\Theta'$, inducing bias in

---

**Algorithm 1:** Sample-Matched PLR (SAMPLR)

Randomly initialize policy $\pi(\phi)$, an empty level buffer $\Lambda$ of size $K$, and belief model $\mathcal{B}(s_t|\tau)$.

**while** *not converged* **do**

    Sample replay-decision Bernoulli, $d \sim \overline{P}_D(d)$

    **if** $d = 0$ or $|\Lambda| = 0$ **then**

        Sample level $\theta$ from level generator

        Collect $\pi$'s trajectory $\tau$ on $\theta$, with a stop-gradient $\phi_\perp$

    **else**

        Use PLR to sample a replay level from the level store, $\theta \sim \Lambda$

        Collect fictitious trajectory $\tau'$ on $\theta$, based on $s'_t \sim \mathcal{B}$

        Update $\pi$ with rewards $\boldsymbol{R}(\tau')$

    **end**

    Compute PLR score, $S = \mathbf{score}(\tau', \pi)$

    Update $\Lambda$ with $\theta$ using score $S$

**end**

---

the resulting decisions. Ideally, we can eliminate this bias, ensuring the resulting policy makes optimal decisions with respect to the ground-truth utility function, conditioned on the current trajectory:

$$\overline{U}(\pi|\tau) = \mathbb{E}_{\theta' \sim \overline{P}(\theta'|\tau)} \left[ \overline{U}(\pi|\tau, \theta') \right]. \qquad (4)$$

A naive solution for grounding is to simply exclude $\Theta'$ from the set of environment parameters under curriculum control. That is, for each environment setting proposed by the curriculum, we

resample $\theta' \sim \overline{P}$. We refer to this approach as *naive grounding*. Naive grounding forces the expected reward and next state under each transition at the current AOH $\tau$ to match that under $\overline{P}$. Thus, optimal policies under naive grounding must be optimal with respect to the ground-truth distribution over $\theta'$.

While technically simple, naive grounding suffers from lack of control over $\Theta'$. This limitation is of no concern when the value of $\Theta'$ does not alter the distribution of $\tau$ until the terminal transition, e.g. when $\Theta'$ is the correct choice in a binary choice task, thereby only influencing the final, sparse reward when the right choice is made. In fact, our initial experiment in Section 6 shows naive grounding performs well in such cases. However, when the value of $\Theta'$ changes the distribution of $\tau$ before the terminal transition, the agent may benefit from a curriculum that actively samples levels which promote learning robust behaviors under unlikely events. Enabling the full benefits of the curriculum in such cases requires the curriculum to selectively sample values of $\Theta'$. Instead of naive grounding, we aim to ground only the policy updates, allowing the curriculum to bias the training distribution. This can be accomplished by optimizing the following objective:

$$\overline{U}_{\mathcal{D}}(\pi) = \mathbb{E}_{\tau \sim \mathcal{D}} \left[ \overline{U}(\pi|\tau) \right]. \tag{5}$$

To achieve this, we replace the reward $r_t$ and next state $s_{t+1}$ with counterfactual values that would be experienced if $\theta'$ were consistent with $\tau$ and $\overline{P}$, so that $\theta' \sim \overline{P}(\theta'|\tau)$. This substitution occurs by simulating a *fictitious transition*, where the fictitious state is sampled as $s'_t \sim \mathcal{B}(s'_t|\tau)$, the action as $a_t \sim \pi(\cdot|\tau)$ (as per usual), the fictitious next state as $s'_{t+1} = \mathcal{T}(s'_t, a_t)$, and the fictitious reward as $r'_t = \mathcal{R}(s'_{t+1})$. The belief model $\mathcal{B}(s'_t|\tau)$ is the ground-truth posterior of the current state given $\tau$:

$$\mathcal{B}(s_t|\tau) = \sum_{\theta'} \overline{P}(s_t|\tau, \theta')\overline{P}(\theta'|\tau). \tag{6}$$

Fictitious transitions, summarized in Figure 2, ground the observed rewards and state transitions to $\overline{P}$. Should training on these transitions lead to an optimal policy over $\Theta$, this policy will also be optimal with respect to $\overline{P}$. We prove this property in Section 5. Fictitious transitions thus provide the benefit of naive grounding without giving up curriculum control over $\Theta'$.

In general, we implement $\mathcal{B}$ as follows: Given $\overline{P}(\Theta')$ as a prior, we model the posterior $\overline{P}(\theta'|\tau)$ with Bayesian inference. The posterior could be learned via supervised learning with trajectories collected from the environment for a representative selection of $\theta'$. Further, we may only have limited access to $\overline{P}(\Theta)$ throughout training, for example, if sampling $\overline{P}(\Theta)$ is costly. In this case, we can learn an estimate $\hat{P}(\Theta')$ from samples we do collect from $\overline{P}(\Theta)$, which can occur online. We can then use $\hat{P}(\Theta')$ to inform the belief model.

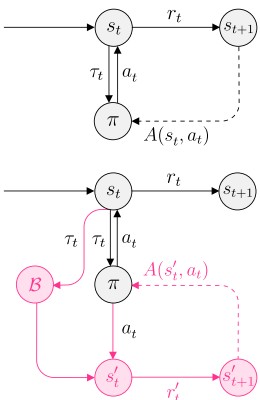

Figure 2: A standard RL transition (top) and a fictitious transition used by SAMPLR (bottom). $A$ is the advantage function.

SAMPLR, summarized in Algorithm 1, incorporates this fictitious transition into PLR$^\perp$ by replacing the transitions experienced in replay levels sampled by PLR$^\perp$ with their fictitious counterparts, as PLR$^\perp$ only trains on these trajectories. PLR$^\perp$ uses PPO with the Generalized Advantage Estimator [GAE, 41] as the base RL algorithm, where both advantage estimates and value losses can be written in terms of one-step TD errors $\delta_t$ at time $t$. Training on fictitious transitions then amounts to computing these TD errors with fictitious states and rewards: $\delta_t = r'_t + V(s'_t) - V(s'_{t+1})$.

Importantly, because PLR$^\perp$ provably leads to policies that minimize worst-case regret over all $\theta$ at NE, SAMPLR enjoys the same property for $\theta \sim \overline{P}(\Theta)$. A proof of this fact is provided in Section 5.

Applying SAMPLR requires two key assumptions: First, the simulator can be reset to a specific state, which is often true, as RL largely occurs in resettable simulators or those that can be made to do so. When a resettable simulator is not available, a possible solution is to learn a model of the environment which we leave for future work. Second, we have knowledge of $\overline{P}(\Theta')$. Often, we know $\overline{P}$ *a priori*, e.g. via empirical data or as part of the domain specification, as in games of chance.

## 5 The Grounded Optimality of SAMPLR

Training on fictitious transitions is a method for learning an optimal policy with respect to the ground-truth utility function $\overline{U}_{\mathcal{D}}(\pi)$ over the distribution $\mathcal{D}$ of training trajectories $\tau$, defined in Equation 5.

When $\mathcal{D}$ corresponds to the distribution of trajectories on levels $\theta \sim \overline{P}(\Theta)$, $\overline{U}_{\mathcal{D}}(\pi)$ reduces to the ground-truth utility function, $\overline{U}(\pi)$. For any UED method, our approach ensures that, in equilibrium, the resulting policy is Bayes-optimal with respect to $\overline{P}(\Theta)$ for all trajectories in the support of $\mathcal{D}$.

**Remark 1.** *If $\pi^*$ is optimal with respect to the ground-truth utility function $\overline{U}_{\mathcal{D}}(\pi)$ then it is optimal with respect to the ground-truth distribution $\overline{P}(\Theta)$ of environment parameters on the support of $\mathcal{D}$.*

*Proof.* By definition we have $\pi^* \in \arg\max_{\pi \in \Pi}\{\overline{U}_{\mathcal{D}}(\pi)\} = \arg\max_{\pi \in \Pi}\{\mathbb{E}_{\tau \sim \mathcal{D}}\left[\overline{U}(\pi|\tau)\right]\}$. Since $\pi$ can condition on the initial trajectory $\tau$, the action selected after each trajectory can be independently optimized. Therefore, for all $\tau \in \mathcal{D}$, $\pi^* \in \arg\max_{\pi \in \Pi}\{\overline{U}(\pi|\tau)\}$ implying that $\pi^*$ is the optimal policy maximizing $\overline{U}(\pi|\tau)$. $\qquad\square$

Thus, assuming the base RL algorithm finds Bayes-optimal policies, a UED method that optimizes the ground-truth utility function, as done by SAMPLR, results in Bayes-optimal performance over the ground-truth distribution. If the UED method maximizes worst-case regret, we can prove an even stronger property we call *robust $\epsilon$-Bayes optimality*.

Let $\overline{U}_{\theta}(\pi)$ be the ground-truth utility function for $\pi$ on the distribution $\mathcal{D}_{\theta}^{\pi}$ of initial trajectories sampled from level $\theta$, so that $\overline{U}_{\theta}(\pi) = \overline{U}_{\mathcal{D}_{\theta}^{\pi}}(\pi)$. Given a policy $\overline{\pi}$ maximizing $\overline{U}_{\theta}(\pi)$, we say that $\overline{\pi}$ is robustly $\epsilon$-Bayes optimal iff for all $\theta$ in the domain of $\overline{P}(\Theta)$ and all $\pi'$, we have

$$\overline{U}_{\theta}(\overline{\pi}) \geq \overline{U}_{\theta}(\pi') - \epsilon.$$

Note how this property differs from being simply $\epsilon$-Bayes optimal, which would only imply that

$$\overline{U}(\overline{\pi}) \geq \overline{U}(\pi') - \epsilon.$$

Robust $\epsilon$-Bayes optimality requires $\overline{\pi}$ to be $\epsilon$-optimal on all levels $\theta$ in the support of the ground-truth distribution, even those rarely sampled under $\overline{P}(\Theta)$. We will show that at $\epsilon$-Nash equilibrium, SAMPLR results in a robustly $\epsilon$-Bayes optimal policy for the ground-truth utility function $\overline{U}_{\theta}(\pi)$. In contrast, training directly on levels $\theta \sim \overline{P}(\Theta)$ results in a policy that is only $\epsilon$-Bayes optimal.

**Theorem 1.** *If $\pi^*$ is $\epsilon$-Bayes optimal with respect to $\overline{U}_{\widehat{\mathcal{D}}}(\pi)$ for the distribution $\widehat{\mathcal{D}}$ of trajectories sampled under $\pi$ over levels maximizing the worst-case regret of $\pi$, as occurs under SAMPLR, then $\pi^*$ is robustly $\epsilon$-Bayes optimal with respect to the ground-truth utility function, $\overline{U}(\pi)$.*

*Proof.* Let $\pi^*$ be $\epsilon$-optimal with respect to $\overline{U}_{\widehat{\mathcal{D}}}(\pi)$ where $\widehat{\mathcal{D}}$ is the trajectory distribution under $\pi$ on levels maximizing the worst-case regret of $\pi$. Let $\overline{\pi}^*$ be an optimal grounded policy. Then for any $\theta$,

$$\overline{U}_{\theta}(\overline{\pi}^*) - \overline{U}_{\theta}(\pi^*) \leq \overline{U}_{\widehat{\mathcal{D}}}(\overline{\pi}^*) - \overline{U}_{\widehat{\mathcal{D}}}(\pi^*) \leq \epsilon \qquad (7)$$

The first inequality follows from $\widehat{\mathcal{D}}$ being trajectories from levels that maximize worst-case regret with respect to $\pi^*$, and the second follows from $\pi^*$ being $\epsilon$-optimal on $\overline{U}_{\widehat{\mathcal{D}}}(\pi)$. Rearranging terms gives the desired condition. $\qquad\square$

## 6 Experiments

Our experiments first focus on a discrete, stochastic binary choice task, with which we validate our theoretical conclusions by demonstrating that CICS can indeed lead to suboptimal policies. Moreover, we show that naive grounding suffices for learning robustly optimal policies in this setting. However, as we have argued, naive grounding gives up control of the aleatoric parameters $\Theta'$ and thus lacks the ability to actively sample scenarios helpful for learning robust behaviors—especially important when such scenarios are infrequent under the ground-truth distribution $\overline{P}(\Theta)$. SAMPLR induces potentially biased curricula, but retains optimality under $\overline{P}(\Theta)$ by matching transitions under $P(\Theta')$ with those under $\overline{P}(\Theta')$. We assess the effectiveness of this approach in our second experimental domain, based on the introductory example of driving icy roads. In this continuous-control driving domain, we seek to validate whether SAMPLR does in fact learn more robust policies that transfer to tail cases under $\overline{P}(\Theta')$, while retaining high expected performance on the whole distribution $\overline{P}(\Theta')$.

All agents are trained using PPO [42] with the best hyperparameters found via grid search using a set of validation levels. We provide extended descriptions of both environments alongside the full details of our architecture and hyperparameter choices in Appendix C.

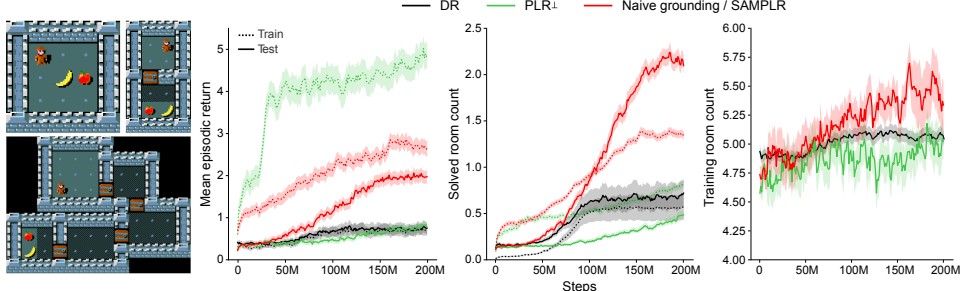

Figure 3: Left: Example Stochastic Fruit Choice levels. The plots show mean and standard error (over 10 runs) of episodic returns (left); room count of solved levels (middle), during training (dotted lines) and test on the ground-truth distribution (solid lines), for $q = 0.7$; and the room count of levels presented at training (right).

## 6.1 Stochastic Fruit Choice

We aim to demonstrate the phenomenon of CICS in Stochastic Fruit Choice, a binary choice task, where the aleatoric parameter determines the correct choice. This task requires the agent to traverse up to eight rooms, and in the final room, decide to eat either the apple or banana. The correct choice $\theta'$ is fixed for each level, but hidden from the agent. Optimal decision-making depends on the ground-truth distribution over the correct fruit, $\overline{P}(\Theta')$. This task benefits from a curriculum over the number of rooms, but a curriculum that selectively samples over both room layout and correct fruit choice can lead to suboptimal policies. Figure 3 shows example levels from this environment.

This domain presents a hard exploration challenge for RL agents, requiring robust navigation across multiple rooms. Further, this environment is built on top of MiniHack [39], enabling integration of select game dynamics from the NetHack Learning Environment [23], which the agent must master to succeed: To go from one room to the next, the agent needs to learn to kick the locked door until it opens. Upon reaching the final room, the agent must then apply the eat action on the correct fruit.

Let $\pi_A$ be the policy that always chooses the apple, and $\pi_B$, the banana. If the probability that the goal is the apple is $\overline{P}(A) = q$, then the expected return is $R_A q$ under $\pi_A$ and $R_B(1 - q)$ under $\pi_B$. The optimal policy is $\pi_A$ when $q > R_B/(R_A + R_B)$, and $\pi_B$ otherwise. Domain randomization (DR), which directly samples each level $\theta \sim \overline{P}(\theta)$, optimizes for the correct ground-truth $\overline{P}(\Theta')$, but will predictably struggle to solve the exploration challenge. PLR$^\perp$ may induce curricula easing the exploration problem, but can be expected make the correct fruit choice oscillate throughout training to maximize regret, leading to problematic CICS.

We set $R_A = 3$, $R_B = 10$, and $q = 0.7$, making $\pi_B$ optimal with an expected return of $3.0$. We compare the train and test performance of agents trained with DR, PLR$^\perp$, and PLR$^\perp$ with naive grounding over 200M training steps in Figure 3. In this domain, SAMPLR reduces to naive grounding, as $\theta'$ only effects the reward of a terminal transition, making fictitious transitions equivalent to real transitions for all intermediate time steps. We see that DR struggles to learn an effective policy, plateauing at a mean return around $1.0$, while PLR$^\perp$ performs the worst. Figure 6 in Appendix B shows that the PLR$^\perp$ curriculum exhibits much higher variance in $q$, rapidly switching the optimal choice of fruit to satisfy its regret-maximizing incentive, making learning more difficult. In contrast, PLR$^\perp$ with naive grounding constrains $q = 0.7$, while still exploiting a curriculum over an increasing number of rooms, as visible in Figure 6. This grounded curriculum results in a policy that solves more complex room layouts at test time. Figures 5 and 6 in Appendix B additionally show how the SAMPLR agent's choices converge to $\pi_B$ and how the size of SAMPLR's improvement varies under alternative choices of $q$ in $\{0.5, 0.3\}$.

## 6.2 Zero-Shot Driving Formula 1 Tracks with Black Ice

We now turn to a domain where the aleatoric parameters influence the distribution of $\tau_t$ at each $t$, thereby creating opportunities for a curriculum to actively sample specific $\theta'$ to promote learning on biased distributions of $\tau_t$. We base this domain on the black ice driving scenario from the introduction of this paper, by modifying the CarRacingBezier environment in [20]. In our version, each track tile has black ice with probability $q$, in which case its friction coefficient is 0, making acceleration and

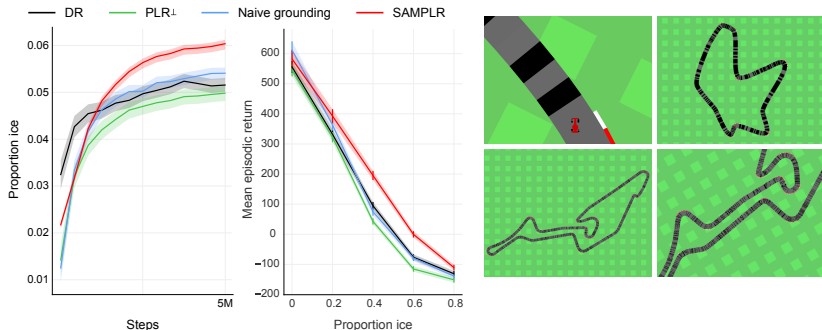

Figure 4: Charts show mean and standard error (over 10 runs) of fraction of visited tiles with ice during training (left) and zero-shot performance on the full Formula 1 benchmark as a function of ice rate (right). Top row screenshots show the agent approaching black ice ($q = 0.4$) and an example training track ($q = 0.6$). Bottom row shows a Formula 1 track ($q = 0.2$) at two zoom scales.

braking impossible. This task is especially difficult, since the agent cannot see black ice in its pixel observations. Figure 4 shows example tracks with ice rendered for illustration purposes. The episodic returns scale linearly with how much of the track is driven and how quickly this is accomplished. As success requires learning to navigate the challenging dynamics over ice patches, a curriculum targeting more difficult ice configurations should lead to policies more robust to black ice. Here, the ground-truth distribution $\overline{P}(\Theta')$ models the realistic assumption that most days see little to no ice. We therefore model the probability of ice per tile as $q \sim \text{Beta}(\alpha, \beta)$, where $\alpha = 1$, $\beta = 15$.

We test the hypothesis that SAMPLR's regret-maximizing curriculum results in policies that preserve optimal performance on the ground-truth distribution $\overline{P}(\Theta')$, while being more robust to tail cases compared to DR and PLR$^\perp$ with naive grounding. We expect standard PLR$^\perp$ to underperform all methods due to CICS, leading to policies that are either too pessimistic or too optimistic with respect

Table 1: Icy F1 returns, mean ± standard error over 10 runs.

| Condition | DR | PLR | Naive | SAMPLR |
|---|---|---|---|---|
| *Ground truth* | | | | |
| $q \sim \text{Beta}(1, 15)$ | $581 \pm 23$ | $543 \pm 21$ | $\mathbf{618 \pm 6}$ | $\mathbf{616 \pm 6}$ |
| *Zero-shot* | | | | |
| $q = 0.2$ | $\mathbf{332 \pm 63}$ | $\mathbf{323 \pm 60}$ | $363 \pm 15$ | $\mathbf{393 \pm 13}$ |
| $q = 0.4$ | $94.7 \pm 41$ | $43 \pm 38$ | $75 \pm 39$ | $\mathbf{195 \pm 11}$ |
| $q = 0.6$ | $-76.3 \pm 24$ | $-115 \pm 12$ | $-79 \pm 25$ | $\mathbf{-1 \pm 17}$ |
| $q = 0.8$ | $-131.1 \pm 11$ | $-151 \pm 6.0$ | $-139 \pm 9$ | $\mathbf{-111 \pm 7}$ |

to the amount of ice. These baselines provide the controls needed to distinguish performance changes due to the two grounding approaches and those due to the underlying curriculum learning method.

We train agents with each method for 5M and test zero-shot generalization performance on the Formula 1 (F1) tracks from the CarRacingF1 benchmark, extended to allow each track segment to have black ice with probability $q$ in $\{0.0, 0.2, 0.4, 0.6, 0.8\}$. These test tracks are significantly longer and more complex than those seen at training, as well as having a higher rate of black ice.

To implement SAMPLR's belief model, we use a second simulator as a perfect model of the environment. At each time step, this second simulator, which we refer to as the *fictitious simulator*, resets to the exact physics state of the primary simulator, and its icy tiles are resampled according to the exact posterior over the aleatoric parameter $q = \theta'$, such that $\theta' \sim \overline{P}(\theta'|\tau)$, ensuring the future uncertainty is consistent with the past. The agent decides on action $a_t$ based on the current real observation $o_t$, and observes the fictitious return $r'_t$ and next state $s'_{t+1}$ determined by the fictitious simulator after applying $a_t$ in state $s'_t \sim \overline{P}(s'_t|\tau, \theta')$. This dual simulator arrangement, fully detailed in Appendix A.2, allows us to measure the impact of training on fictitious transitions independently of the efficacy of a model-based RL approach. Further, as the training environment in RL is most often simulation (e.g. in sim2real), this approach is widely applicable.

SAMPLR outperforms all baselines in zero-shot transfer to higher ice rates on the full F1 benchmark and attains a statistically significant improvement at $p < 0.001$ when transferring to $q = 0.4$ and $q = 0.6$, and $p < 0.05$ when $q = 0.8$. Importantly, SAMPLR outperforms PLR$^\perp$ with naive grounding, indicating that SAMPLR exploits specific settings of $\Theta'$ to better robustify the agent against rare icy conditions in the tail of $\overline{P}(\Theta')$. Indeed, Figure 4 shows that on average, SAMPLR exposes the agent to more ice per track tile driven, while PLR$^\perp$ underexposes the agent to ice compared to DR and naive grounding, suggesting that under PLR$^\perp$ agents attain higher regret on ice-free tracks—a likely outcome as ice-free tracks are easier to drive and lead to returns, with

which regret scales. Unfortunately, this results in PLR$^\perp$ being the worst out of all methods on the ground-truth distribution. SAMPLR and naive grounding avoid this issue by explicitly matching transitions to those under $\overline{P}$ at $\tau$. As reported in Table 1, SAMPLR matches the baselines in mean performance across all F1 tracks under $\overline{P}(\Theta')$, indicating that despite actively sampling challenging $\theta'$, it preserves performance under $\overline{P}(\Theta')$, i.e. the agent does not become overly cautious.

## 7 Related Work

The mismatch between training and testing distributions of input features is referred to as *covariate shift*, and has long served as a fundamental problem for the machine learning community. Covariate shifts have been extensively studied in supervised learning [51, 18, 5, 2]. In RL, prior works have largely focused on covariate shifts due to training on off-policy data [47, 37, 11, 14, 13, 49] including the important case of learning from demonstrations [33, 36]. Recent work also aimed to learn invariant representations robust to covariate shifts [56, 57]. More generally, CICS is a form of sample-selection bias [15]. Previous methods like OFFER [7] considered correcting biased transitions via importance sampling [46] when optimizing for expected return on a single environment setting, rather than robust policies over all environments settings. We believe our work provides the first general formalization and solution strategy addressing curriculum-induced covariate shifts (CICS) for RL.

The importance of addressing CICS is highlighted by recent results showing curricula to be essential for training RL agents across many of the most challenging domains, including combinatorial gridworlds [58], Go [43], StarCraft II [52], and achieving comprehensive task mastery in open-ended environments [44]. While this work focuses on PLR$^\perp$, other methods include minimax adversarial curricula [32, 53, 54] and curricula based on changes in return [25, 34]. Curriculum methods have also been studied in goal-conditioned RL [12, 6, 45, 29], though CICS does not occur here as goals are observed by the agent. Lastly, domain randomization [DR, 38, 31] can be seen as a degenerate form of UED, and curriculum-based extensions of DR have also been studied [19, 50].

Prior work has also investigated methods for learning Bayes optimal policies under uncertainty about the task [59, 30], based on the framework of Bayes-adaptive MDPs (BAMDPs) [3, 10]. In this setting, the agent can adapt to an unknown MDP over several episodes by acting to reduce its uncertainty about the identity of the MDP. In contrast, SAMPLR learns a robustly Bayes-optimal policy for zero-shot transfer. Further unlike these works, our setting assumes the distribution of some aleatoric parameters is biased during training, which would bias the *a posteriori* uncertainty estimates with respect to the ground-truth distribution when optimizing for the BAMDP objective. Instead, SAMPLR proposes a means to correct for this bias assuming knowledge of the true environment parameters, to which we can often safely assume access in curriculum learning.

Deeply related, Off-Belief Learning [OBL, 16] trains cooperative agents in self-play using fictitious transitions assuming all past actions of co-players follow a base policy, e.g. a uniformly random one. Enforcing this assumption prevents agents from developing conventions that communicate private information to co-players via arbitrary action sequences. Such conventions hinder coordination with independently trained agents or, importantly, humans. SAMPLR can be viewed as adapting OBL to single-agent curriculum learning, where a co-player sets the environment parameters at the start of each episode (see Appendix D). This connection highlights how single-agent curriculum learning is inherently a multi-agent problem, and thus problems afflicting multi-agent learning also surface in this setting; moreover, methods addressing such issues in one setting can then be adapted to the other.

## 8 Conclusion

This work characterized how curriculum-induced covariate shifts (CICS) over aleatoric environment parameters $\Theta'$ can lead to suboptimal policies under the ground-truth distribution over these parameters, $\overline{P}(\Theta')$. We introduced a general strategy for correcting CICS, by training the agent on fictitious rewards and next states whose distribution is guaranteed to match what would be experienced under $\overline{P}(\Theta')$. Our method SAMPLR augments PLR$^\perp$ with this correction. By training on fictitious transitions, SAMPLR actively samples specific values of $\theta'$ that induce trajectories with greater learning potential, while still grounding the training data to $\overline{P}(\Theta')$. Crucially, our experiments in challenging environments with aleatoric uncertainty showed that SAMPLR produces robust policies outperforming those trained with competing baselines that do not correct for CICS.

A core assumption made by SAMPLR and all other UED methods is the ability to reset the environment to arbitrary configurations of some set of free parameters. While such resets can be difficult or impossible to perform in real world environments, in practice, this assumption is nearly always satisfied, as RL training largely occurs under a sim2real paradigm due to the additional costs of training in the wild. Most RL simulators can either be directly reset to specific environment configurations or be straightforwardly made to do so. SAMPLR thus provides a means to more fully exploit the affordances of a simulator to produce more robust policies: Policies trained with SAMPLR retain optimality when transferred to the ground-truth distribution of aleatoric parameters in the real environment—a crucial property not satisfied by prior UED methods. Importantly, the approach based on fictitious transitions used by SAMPLR can, in principle, be generally applied to prior UED methods to provide them with this desirable property.

## Acknowledgments and Disclosure of Funding

We thank Robert Kirk and Akbir Khan for providing useful feedback on earlier drafts of this work. Further, we are grateful to our anonymous reviewers for their valuable feedback. MJ is supported by the FAIR PhD program at Meta AI. This work was funded by Meta.

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
