# A Algorithms

## A.1 Robust Prioritized Level Replay

We briefly describe the PLR$^\perp$ algorithm and provide its pseudocode for completeness. For a detailed discussion of PLR$^\perp$ and its theoretical guarantees, we point the reader to [20].

For a given UPOMDP, PLR$^\perp$ induces a curriculum by using random search over the set of possible environment configurations $\theta$ to find the high regret levels for training. In order to estimate regret for level $\theta$, PLR$^\perp$ uses an estimator **score**$(\tau, \pi)$ that is a function of the last trajectory on that level $\tau$ and the agent's policy $\pi$ that generated $\tau$. In practice these estimates are based on variations of the time-averaged value loss over $\tau$. At the start of each episode with probability $1 - p$, PLR$^\perp$ samples a new level $\theta$ from the training distribution $\overline{P}$, and uses the ensuing episode purely for evaluation, without training the agent on the collected experiences. It maintains a level buffer $\mathbf{\Lambda}$ of the top-$K$ highest regret levels found throughout training. With probability $p$, episodes are used for training on levels sampled from this set, according to the *replay distribution* $P_{\text{replay}}$. Given staleness coefficient $\rho$, temperature $\beta$, a prioritization function $h$ (e.g. rank), level buffer scores $S$, level buffer timestamps $C$, and the current episode count $c$ (i.e. current timestamp), $P_{\text{replay}}$ is defined as

$$P_{\text{replay}} = (1 - \rho) \cdot P_S + \rho \cdot P_C,$$
$$P_S = \frac{h(S_i)^{1/\beta}}{\sum_j h(S_j)^{1/\beta}},$$
$$P_C = \frac{c - C_i}{\sum_{C_j \in C} c - C_j}.$$

The first component $P_S$ weighs each level based on its estimated regret, and the second, $P_C$, based on the age of the regret estimate. The staleness term mitigates regret values drifting off policy during training. The staleness coefficient $\rho$ controls the contribution of each component. This process is summarized in Algorithm 2, and the details of how PLR$^\perp$ updates the top-$K$ high regret levels in $\mathbf{\Lambda}$, in Algorithm 3.

---

**Algorithm 2:** Robust PLR (PLR$^\perp$)

---

Randomly initialize policy $\pi(\phi)$ and an empty level buffer, $\mathbf{\Lambda}$ of size $K$.
**while** *not converged* **do**
    Sample replay-decision Bernoulli,
      $d \sim P_D(d)$
    **if** $d = 0$ **then**
        Sample level $\theta$ from level generator
        Collect $\pi$'s trajectory $\tau$ on $\theta$, with a
            stop-gradient $\phi_\perp$
    **else**
        Use PLR to sample a replay level from
          the level store, $\theta \sim \mathbf{\Lambda}$
        Collect policy trajectory $\tau$ on $\theta$ and
          update $\pi$ with rewards $\boldsymbol{R}(\tau)$
    **end**
    Compute PLR score, $S = $ **score**$(\tau, \pi)$
    Update $\mathbf{\Lambda}$ with $\theta$ using score $S$
**end**

---

**Algorithm 3:** PLR level-buffer update rule

---

**Input:** Level buffer $\mathbf{\Lambda}$ of size $K$ with scores $S$
         and timestamps $C$; level $\theta$; level score
         $S_\theta$; and current episode count $c$
**if** $|\mathbf{\Lambda}| < K$ **then**
    Insert $\theta$ into $\mathbf{\Lambda}$, and set $S(\theta) = S_\theta$,
      $C(\theta) = c$
**else**
    Find level with minimal support,
      $\theta_{\min} = \arg\min_\theta P_{\text{replay}}(\theta)$
    **if** $S(\theta_{\min}) < S_\theta$ **then**
        Remove $\theta_{\min}$ from $\mathbf{\Lambda}$
        Insert $\theta$ into $\mathbf{\Lambda}$, and set $S(\theta) = S_\theta$,
          $C(\theta) = c$
        Update $P_{\text{replay}}$ with latest scores $S$ and
          timestamps $C$
    **end**
**end**

---

## A.2 SAMPLR Implementation Details

In the car racing environment, the aleatoric parameters $\theta'$ determine whether each track tile contains black ice. Thus, the training distribution $P(\Theta')$ directly impacts the distribution over $\tau$. In order to correct for the biased trajectories $\tau$ generated under its minimax regret curriculum, we must train the policy to maximize the ground-truth utility function conditioned on $\tau$, $\overline{U}(\pi|\tau)$. SAMPLR

accomplishes this by training the policy on fictitious transitions that replace the real transitions observed. Each fictitious transition corresponds to the reward $r'_t$ and $s'_{t+1}$ that would be observed if the agent's action were performed in a level such that $\theta' \sim \overline{P}(\theta'|\tau)$. By training the agent on a POMDP whose future evolution conditioned on $\tau$ is consistent with $\overline{P}$, we ensure any optimal policy produced under the biased training distribution will also be optimal under $\overline{P}(\Theta')$.

Recall from Equation 6 that $\mathcal{B}(s'_t|\tau) = \sum_{\theta'} \overline{P}(s'_t|\tau, \theta')\overline{P}(\theta'|\tau)$. We can thus sample a fictitious state $s'_t$ according to $\mathcal{B}$ by first sampling $\theta' \sim \overline{P}(\theta'|\tau)$, and then $s'_t \sim \overline{P}(s'_t|\tau, \theta')$. We implement SAMPLR for this domain by assuming perfect models for both the posterior $P(\theta'|\tau)$ and $\overline{P}(s'_t|\tau, \theta')$.

Simulating a perfect posterior over $\theta'$ is especially straightforward, as we assume each tile has ice sampled I.I.D. with probability $q \sim \text{Beta}(\alpha, \beta)$, where we make use of the conjugate prior. As $\tau$ contains the entire action-observation history up to the current time, it includes information that can be used to infer how much ice was already seen. In order to simulate a perfect posterior over $\theta'$, we thus track whether each visited track tile has ice and use these counts to update an exact posterior over $q$, equal to $\text{Beta}(\alpha + N_+, \beta + N_-)$, where $N_+$ and $N_-$ correspond to the number of visited tiles with and without ice respectively. We then effectively sample $\theta' \sim \overline{P}(\theta'|\tau)$ by resampling all unvisited tiles from this posterior.

In order to sample from $\overline{P}(s'_t|\tau, \theta')$ and similarly from the grounded transition distribution $\overline{P}(s'_{t+1}|a_t, \tau, \theta')$, we make use of a second simulator we call the *fictitious simulator*, which acts as a perfect model of the environment. We could otherwise learn this model via online or offline supervised learning. Our design choice using a second simulator in place of such a model allows us to cleanly isolate the effects of SAMPLR's correction for CICS from potential errors due to the inherent difficulties of model-based RL.

Let us denote the primary simulator by $\mathcal{E}$, and the fictitious simulator by $\mathcal{E}'$. We ensure that the parameters of both simulators always match for $\theta \in \Theta \setminus \Theta'$. Before each training step, we first set the physics state of $\mathcal{E}'$ to that of $\mathcal{E}$ exactly, ensuring both simulators correspond to the same $s_t$, and then resample $\theta' \sim \overline{P}(\theta'|\tau)$ for the fictitious simulator as described above. We then take the resulting state of $\mathcal{E}'$ as $s'_t$. The agent next samples an action from its policy, $a_t \sim \pi(a_t|s'_t)$. Stepping forward $\mathcal{E}'$ in state $s'_t$ with action $a_t$ then produces a sample of $s'_{t+1}$ from a perfect grounded belief model, along with the associated reward, $r'_t$. Throughout PPO training, the 1-step TD-errors $\delta_t$ for time $t$ are computed using these fictitious transitions. Similarly, the PLR$^\perp$ mechanism underlying SAMPLR estimates regret using $\delta_t$ based on fictitious transitions.

# B  Additional Experimental Results

## B.1  Stochastic Fruit Choice

When the probability of apple being the correct goal is $q = 0.7$, and the payoff for correctly choosing the apple (banana) is 3 (10), the optimal policy in expectation is to always choose the banana, resulting in an expected return of 3.0. Figure 5 shows that both DR and PLR$^\perp$ fail to learn to consistently eat the banana after 200M steps of training. In contrast, SAMPLR learns to always choose the banana when it successfully solves a level.

In additional experiments, we find that the impact of CICS varies with $q$. When $q = 10/13$, the expected returns for the policy always choosing apple ($\pi_B$) equals that for the policy always choosing banana ($\pi_B$). The top row of Figure 6 shows that the negative impact of CICS on PLR$^\perp$ and thus the benefits of SAMPLR diminish the farther $q$ is from this equilibrium value. Intuitively, for $q$ closer to the equilibrium value, smaller covariate shifts suffice to flip the policy, making it easier for PLR$^\perp$ to rapidly oscillate the optimal policy during training. We see in the bottom row of 6 that PLR$^\perp$ indeed produces large adversarial oscillations in $q$. This makes it difficult for the agent to settle on the optimal policy with respect to any ground-truth distribution. In contrast, SAMPLR grounds PLR's otherwise wild shifts in $q$ with respect to its ground-truth value, allowing the agent to learn a well-grounded policy.

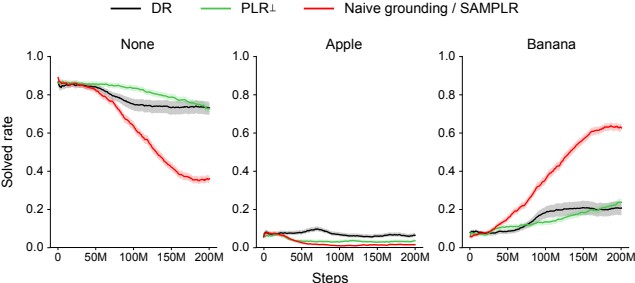

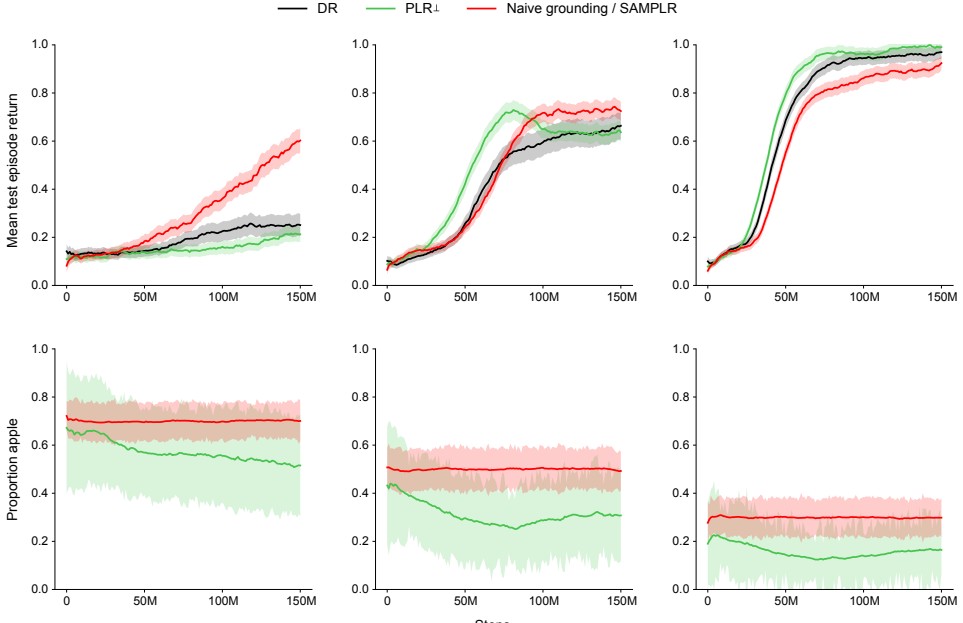

Figure 5: Left: Proportion of training episodes for $q = 0.7$ in which the agent fails to eat any fruit; eats the apple; or eats the banana. Right: Number of rooms in levels during training. Plots show mean and standard error of 10 runs.

Figure 6: Top: Mean and standard error of episodic test returns as the probability $q$ of the apple being the correct choice varies. Bottom: The proportion of training levels chosen by each method where apple is the correct choice. The mean and standard deviation are shown.

## B.2 Car Racing with Black Ice

We see that on a gradient-update basis, SAMPLR and PLR$^\perp$ with naive grounding begin to overtake domain randomization much earlier in training. This is because PLR$^\perp$, does not perform gradient updates after each rollout, but rather only trains on a fraction $p$ of episodes which sample high regret replay levels. In our experiments, we set the replay rate $p = 0.5$, meaning for the same number of training steps, the PLR$^\perp$-based methods only take half as many gradient steps as the DR baseline. Figure 7 shows each method's training and zero-shot transfer performance on validation F1 tracks with $q = 0.2$, as a function of the number of environment steps, while Figure 8 plots the same results as a function of the number of gradient updates performed. On a gradient basis, SAMPLR sees even more pronounced gains in robustness compared to DR—that is, SAMPLR learns more robust policies with less data.

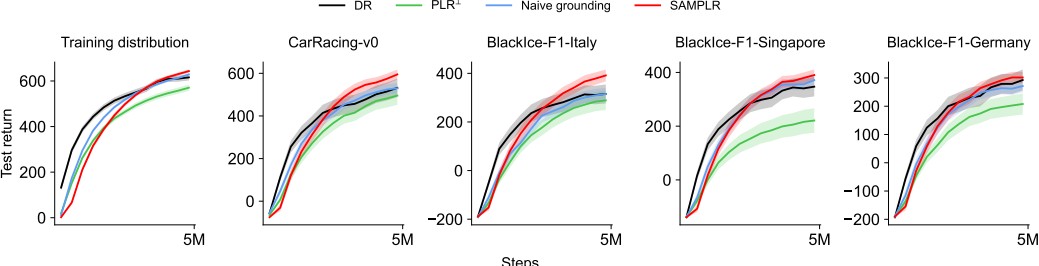

Figure 7: Performance of each method on the training distribution and zero-shot transfer environments with probability of ice per tile $q = 0.2$. These results show mean and standard error of episodic test returns as a function of number of environment steps collected during training.

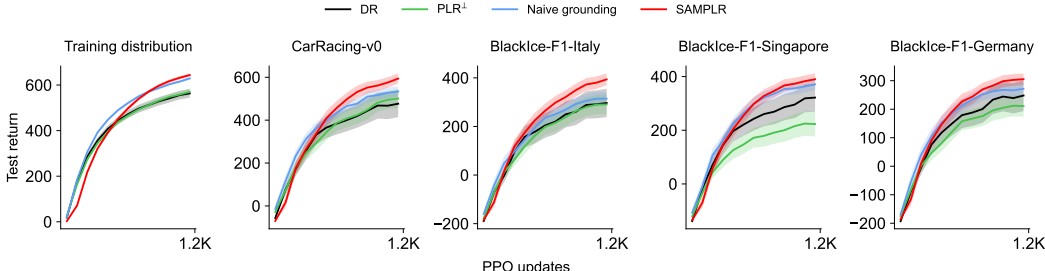

Figure 8: The same performance results reported in Figure 7, but as a function of number of gradient updates performed.

## C    Experimental Details and Hyperparameters

In this section, we provide detailed descriptions of each of the experimental domains featured in this work. In addition, we describe our architecture and hyperparameter choices for each setting. Table 2 summarizes our choice of hyperparameters for each domain.

### C.1    Stochastic Fruit Choice

**Environment details** We make use of MiniHack [39], a library built on top of the NetHack Learning Environment [NLE, 23], for creating custom environments in the NetHack runtime. Our Stochastic Fruit Choice environment embeds a stochastic binary choice task within a challenging hard-exploration problem. The agent must navigate through up to eight rooms in each level, and in the final room, choose the correct piece of fruit, either the apple or banana to receive a reward. If the agent eats the wrong fruit for the level, it receives a reward of $0$. With probability $q$, the apple is the correct fruit to eat. Eating either fruit terminates the episode. The episode also terminates once the budget of $250$ steps is reached. Notably, passage into adjacent rooms requires first kicking down a locked door. As per NLE game dynamics, locked doors may require a random number of kicks before they give way. To complicate the learning of this kicking skill, kicking the stone walls of the room will lower the agent's health points; multiple misguided kicks can then lead to the agent dying, ending the episode.

The agent's observation consists of two primary elements: The nethack `glyph` and `blstats` tensors. The `glyph` tensor represents a 2D symbolic observation of the dungeon. This glyph tensor contains a $21 \times 79$ window of glyph identifiers, which can each be one of the $5991$ possible glyphs in NetHack, which represent monsters, items, environment features, and other game entities. The `blstats` vector contains character-centric values, such as the agent's coordinates and the information in the "bottom-line stats" of the game, such as the agent's health stats, attribute levels, armor class, and experience points.

The action space includes the eight navigational actions, corresponding to moving toward each cell in the agent's Moore neighborhood, in addition to two additional actions for kicking (doors) and eating (apples and bananas).

**Architecture** We make use of the same agent architecture from [23]. The policy applies a ConvNet to all visible glyph embeddings and a separate ConvNet to a $9 \times 9$ egocentric crop around the agent—which was found to improve generalization—producing two latent vectors. These are then concatenated with an MLP encoding of the `blstats` vector, the resulting vector is further processed by an MLP layer, and finally, input through an LSTM to produce the action distribution. We used the policy architecture provided in https://github.com/facebookresearch/nle.

**Choice of hyperparameters** Our choice of PPO hyperparameters, shared across all methods, was based on a grid search, in which we train agents with domain randomization on a $15 \times 15$ maze, in which the goal location and initial agent location, along with up to 50 walls, are randomly placed. For each setting, we average results over 3 training runs. We chose this environment to perform the grid search, as it allows for significantly faster training than the multi-room environment featured in our main experiments. Specifically, we swept over the following hyperparameter values: number of PPO epochs in $\{5, 20\}$, number of PPO minibatches in $\{1, 4\}$, PPO clip parameter in $\{0.1, 0.2\}$, learning rate in $\{1e\text{-}3, 1e\text{-}4\}$, and discount factor $\gamma$ in $\{0.99, 0.995\}$. Fixing these hyperparameters to the best setting found, we then performed a separate grid search over PLR's replay rate $p$ in $\{0.5, 0.95\}$ and replay buffer size in $\{4000, 5000, 8000\}$, evaluating settings based on evaluation levels sampled via domain randomization after 50M steps of training.

**Compute** All agents in the Stochastic Fruit Choice environment were trained using Tesla V100 GPUs and Intel Xeon E5-2698 v4 CPUs. DR, PLR$^{\perp}$, and naive grounding all required approximately 24 hours to complete the 50M training steps for each run of the hyperparameter sweep and 192 hours to complete the 200 million training steps for each full multi-room run. In total, our experimental results, including hyperparameter sweeps, required roughly 8,500 hours (around 350 days) of training.

## C.2   Car Racing with Black Ice

**Environment details** This environment extends the CarRacingBezier environment from [20] to optionally contain black ice on each track tile with probability $q$, where for a given track, $q$ may first be sampled from an arbitrary prior distribution and after which, ice is sampled I.I.D. per tile. It is impossible to accelerate or brake over ice tiles, which have a friction coefficient of 0, making icy settings much more challenging. Like in CarRacingBezier, each track shape is generated by sampling a random set of 12 control points that define a closed Bézier curve. For a track composed of $L$ tiles, the agent receives a reward of $1000/L$ for each tile visited, and a per-step penalty of $-0.1$. Like [20], we adopt the methodology outlined in [24] and do not penalize the agent for driving out of bounds, terminate episodes early when the agent drives too far off the track, and repeat each action for 8 frames. At each time step, the agent receives a $96 \times 96 \times 3$ RGB frame consisting of a local birdseye view of the car on the track and a dashboard showing the agent's latest action and return. Actions take the form of a three-dimensional, continuous vector. Each component represents control values for torque in $[-1.0, 1.0]$, acceleration in $[0.0, 1.0]$, and deceleration in $[0.0, 1.0]$.

**Architecture** We adopt a policy architecture used across several prior works [20, 24, 48], consisting of a stack of 2D convolutions feeding into a fully-connected ReLU layer. The convolutions have square kernels of size 2, 2, 2, 2, 3, 3, output channels of dimension 8, 16, 32, 64, 128, 256, and stride lengths of 2, 2, 2, 2, 1, 1. The resulting 256-dimensional embedding is then fed into alpha, beta, and value heads. The alpha and beta heads are each fully-connected softplus layers, to whose outputs we add 1 to produce the $\alpha$ and $\beta$ parameters for the Beta distribution parameterizing each action dimension (i.e. each action is sampled from Beta$(\alpha, \beta)$ and then translated into the appropriate range). The value head is a fully-connected ReLU layer. All hidden layers are 100-dimensional.

**Choice of hyperparameters** We selected hyperparameters based on evaluation performance over 5 episodes on the Italy, Singapore, and Germany F1 tracks with ice probability per tile fixed to $q = 0.2$, when trained under the ice distribution featured in our main results, where $q \sim$ Beta(1,15). For each setting, we averaged results over 3 runs. PPO hyperparameters, shared across methods, were selected based on the performance of agents trained with domain randomization across settings in a grid search covering learning rate in $\{0.001, 0.0003, 0.0001, 0.00001\}$, number of epochs in $\{3, 8\}$, number of minibatches in $\{2, 4, 16\}$, and value loss coefficient in $\{0.5, 1.0, 2.0\}$. The remaining PPO

hyperparameters, as well as PLR$^\perp$-specific hyperparameters were based on those used in [20], with the exception of a smaller level buffer size, which we found helped improve validation performance. Additionally, for each method, we also swept over the choice of whether to initialize the policy to ensure actions are initially close to zero. Initializing the policy in this way has been shown to reduce variance in performance across seeds [1].

**Compute** All car racing agents were trained using Tesla V100 GPUs and Intel Xeon E5-2698 v4 CPUs. DR, PLR$^\perp$, and naive grounding all required approximately 24 hours to complete the 5M training steps in each experiment run, while SAMPLR required 42 hours. In total, our experimental results, including hyperparameter sweeps, required roughly 11,500 hours (around 480 days) of training.

Table 2: Hyperparameters used for training each method.

| Parameter | Stochastic Fruit Choice | Black-Ice Car Racing |
|---|---|---|
| *PPO* | | |
| $\gamma$ | 0.995 | 0.99 |
| $\lambda_{\text{GAE}}$ | 0.95 | 0.9 |
| PPO rollout length | 256 | 125 |
| PPO epochs | 5 | 3 |
| PPO minibatches per epoch | 1 | 4 |
| PPO clip range | 0.2 | 0.2 |
| PPO number of workers | 32 | 16 |
| Adam learning rate | 1e-4 | 1e-4 |
| Adam $\epsilon$ | 1e-5 | 1e-5 |
| PPO max gradient norm | 0.5 | 0.5 |
| PPO value clipping | yes | no |
| return normalization | no | yes |
| value loss coefficient | 0.5 | 1.0 |
| entropy coefficient | 0.0 | 0.0 |
| | | |
| *PLR$^\perp$* and *SAMPLR* | | |
| Replay rate, $p$ | 0.95 | 0.5 |
| Buffer size, $K$ | 4000 | 500 |
| Scoring function | positive value loss | positive value loss |
| Prioritization | rank | power |
| Temperature, $\beta$ | 0.3 | 1.0 |
| Staleness coefficient, $\rho$ | 0.3 | 0.7 |

# D Connection to Off-Belief Learning

In cooperative multi-agent reinforcement learning (MARL), self-play promotes the formation of cryptic conventions—arbitrary sequences of actions that allow agents to communicate information about the environment state. These conventions are learned jointly among all agents during training, but are arbitrary and hence, indecipherable to independently-trained agents or humans at test time. Crucially, this leads to policies that fail to perform zero-shot coordination [ZSC, 17], where independently-trained agents must cooperate successfully without additional learning steps—a setting known as ad-hoc team play. Off-Belief Learning [OBL; 16] resolves this problem by forcing agents to assume their co-players act according to a fixed, known policy $\pi_0$ until the current time $t$, and optimally afterwards, conditioned on this assumption. If $\pi_0$ is playing uniformly random, this removes the possibility of forming arbitrary conventions.

Formally, let $G$ be a decentralized, partially-observable MDP [Dec-POMDP, 4], with state $s$, joint action $a$, observation function $\mathcal{I}^i(s)$ for each player $i$, and transition function $\mathcal{T}(s, a)$. Let the historical trajectory $\tau = (s_1, a_1, ...a_{t-1}, s_t)$, and the action-observation history (AOH) for agent $i$ be $\tau^i = (\mathcal{I}^i(s_1), a_1, ..., a_{t-1}, \mathcal{I}^i(s_t))$. Further, let $\pi_0$ be an arbitrary policy, such as a uniformly random policy, and $\mathcal{B}_{\pi_0}(\tau|\tau^i) = P(\tau_t|\tau_t^i, \pi_0)$, a belief model predicting the current state, conditioned on the AOH of agent $i$ and the assumption of co-players playing policy $\pi_0$ until the current time $t$, and optimally according to $\pi_1$ from $t$ and beyond. OBL aims to find the policy $\pi_1$ with the optimal, *counter-factual value function*,

$$V^{\pi_0 \to \pi_1}(\tau^i) = \mathbb{E}_{\tau \sim \mathcal{B}_{\pi_0}(\tau^i)}\left[V^{\pi_1}(\tau)\right]. \tag{8}$$

Thus, the agent conditions its policy on the realized AOH $\tau^i$, while optimizing its policy for transition dynamics based on samples from $\mathcal{B}_{\pi_0}$, which are consistent with the assumption that co-players play according to $\pi_0$ until time $t$. Therefore, if $\pi_0$ is a uniformly random policy, $\pi_1$ can no longer benefit from conditioning on the action sequences of its co-players, thereby preventing the formation of cryptic conventions that harm ZSC.

Similarly, in single-agent curriculum learning, we can view the UED teacher as a co-player that performs a series of environment design decisions that defines the environment configuration $\theta$ at the start of the episode, and subsequently performs no-ops for the remainder of the episode. As discussed in Section 3, curriculum-induced covariate shifts (CICS) can cause the final policy to be suboptimal with respect to the ground-truth distribution $\overline{P}$ when the teacher produces a curriculum resulting in the training distribution of aleatoric parameters $P(\Theta')$ deviating from the ground-truth distribution $\overline{P}(\Theta')$. We then see that the fictitious transitions used by SAMPLR are equivalent to those used by OBL, where the belief model $\mathcal{B}$ assumes the teacher makes its design choices such that the resulting distribution of aleatoric parameters $\Theta'$ matches the ground-truth $\overline{P}(\Theta')$. This connection reveals that SAMPLR can be viewed as an adaptation of OBL to the single-agent curriculum learning setting, whereby the UED teacher, which designs the environment configuration, is viewed as the co-player.

# E   Broader Impact Statement

SAMPLR trains policies that are optimal under some ground-truth distribution $\overline{P}$ despite using data sampled from a curriculum distribution that may be biased with respect to $\overline{P}$. However, often in practice, our knowledge of $\overline{P}$ may be based on estimations that are themselves biased. For example, when applying SAMPLR with respect to an estimated ground-truth distribution of user demographics, the resulting policy may still be biased toward some demographics, if the ground-truth estimation is biased. Therefore, when applying SAMPLR, we must still take care to ensure our notion of the ground truth is sound.