# OpenReview forum: "Grounding Aleatoric Uncertainty for Unsupervised Environment Design"
_NeurIPS.cc/2022/Conference — NeurIPS 2022 Accept_

### Official Review · Reviewer_EQPK · 2022-07-04

**Rating:** 6
**Confidence:** 3
**Soundness:** 2 fair
**Presentation:** 2 fair
**Contribution:** 2 fair

**Summary:**

This paper focuses on the unsupervised environment design (UED) framework. They first formalized the covariate shift issue that the training distribution differs from the test distribution in curriculum learning. To address this issue, they proposed a new algorithm SAMPLR, which can preserve optimality on the ground-truth distribution while training under a usefully biased training distribution. Finally, they evaluated SAMPLR on two tasks.

**Questions:**

1. In the related work part, the connection between SAMPLR and Off-Belief Learning in multi-agent RL is established. It is not intuitive to understand this connection. Can the authors elaborate more on this point?

**Ethics Review Area:**

["I don’t know"]

**Limitations:**

The limitations of their algorithm are discussed clearly in sections 4 and 5.

**Strengths And Weaknesses:**

Strengths:
1. This paper studies the covariate shift issue in UED, which is an important problem.
2. The proposed algorithm SAMPLR outperforms baselines on two tasks.

Weakness:
1. I am most concerned about the assumptions required by SAMPLR. First, SAMPLR requires access to a simulator that can be reset to any state. In many RL tasks, the simulator cannot be reset to an arbitrary state. Second, SAMPLR requires knowledge of the ground truth distribution.
2. The experiments are insufficient to support their algorithm. First, they tested the proposed algorithm only on two tasks. I am not sure whether the covariate shift issue is common in curriculum learning and whether SAMPLR can consistently outperform baselines. Second, it seems that the stochastic fruit choice task is a bit too simple. In this task, the proposed algorithm reduces to naïve grounding.

---

> ### Author Response · Authors · 2022-07-30
> **Response from authors, 2/2**
>
> ### Requirement of knowing the ground-truth distribution
> It is true that we must know the ground-truth distribution---though importantly, only over the aleatoric parameters. However, in many cases, we know accurate estimates of this distribution, e.g. based on historical data. In such cases, we can directly apply SAMPLR. In this sense, SAMPLR provides a useful way to incorporate such external knowledge about the ground-truth environment to ensure higher performing policies at deployment in the ground-truth setting, while still training on biased transitions under a curriculum distribution. To our knowledge, SAMPLR is the first method to enable this.
>
> Otherwise, in cases where we do not know the exact ground-truth distribution, we can take rollouts on levels sampled from the deployment distribution in order to estimate the ground-truth distribution over the aleatoric parameters. This can be performed simply via domain randomization of the environment configurations, assuming the simulator generates levels in a way that is consistent with the deployment distribution in reality. Of course, this may not always be the case, and this would be an instance of the sim2real gap---the difference between the simulator and reality---which affects all RL methods that train in simulation, and is not a unique issue with SAMPLR. Alternatively, we can collect data about the deployment domain separately from policy training to estimate the ground-truth distribution offline. This latter approach can be widely applied. In general, estimating the ground-truth distribution is a separate problem. Our problem setting assumes some knowledge of the ground-truth distribution, and given this knowledge, we propose a way to ensure curriculum learning remains optimal for the ground-truth distribution.
>
> We are happy to incorporate this discussion into the updated manuscript if you this provides useful context around the requirement of knowledge of the ground-truth distribution.
>
> ### How common is CICS in curriculum learning?
> As we argue in the paper, CICS can cause curriculum learning to produce suboptimal policies for the ground-truth distribution whenever optimal behavior requires conditioning on the ground-truth distribution over aleatoric parameters. Thus, this problem can surface when performing curriculum learning in any environment requiring decision-making under uncertainty. As curriculum learning becomes more widely used, this situation should become more and more common, as many settings contain sources of aleatoric uncertainty. For a straightforward family of examples, consider games of chance as one might find in a casino. Curriculum learning in such environments can lead to highly biased transitions that do not match the transitions one would experience under the ground-truth distribution for the game, e.g. Blackjack or Poker.
>
> ### More details on connection between SAMPLR and OBL
> Both SAMPLR and OBL use fictitious transitions to force the student policy to optimize its performance under the assumption that a co-player’s previous decisions were based on a policy that satisfies some distributional constraint (where the teacher designing designing the environment is the co-player in the case of SAMPLR). In OBL, the co-players are the other agents in a cooperative multi-agent environment, and the constraint is that the previous actions come from some fixed policy $\pi_0$. If $\pi_0$ is set to the uniform random policy, the learned policies will no longer benefit from conditioning on the past actions of co-players, thereby preventing policies trained in self-play to share information about their private state, e.g. the cards they can see in Hanabi, by encoding this information in arbitrary action sequences. Sharing information in this way is bad, as it will not transfer to independently trained agents that can be co-players at test-time, and OBL thus fixes this. In SAMPLR, the co-player can be viewed as the teacher designing the environment, whose policy decides the specific environment configuration at the start of each episode, and the constraint is that it must choose the aleatoric parameter values according to the ground-truth distribution $\overline{P}$. This prevents the teacher from creating curricula with covariate shifts in the aleatoric parameters, thereby preventing harmful cases of CICS, and ensuring the learned policy is optimal under the ground-truth distribution $\overline{P}$. We have included an extended discussion of OBL and its connection to SAMPLR in Appendix D of the updated manuscript.
>
> Given this additional discussion and clarifications, would you consider increasing your support for our work to "Accept"?

---

> > ### Author Response · Authors · 2022-08-09
> > **Let us know if we have addressed your concerns**
> >
> > Given that we provided quite comprehensive responses to the points raised in your review, could you let us know if this further discussion addresses your concerns? If our rebuttal addressed your concerns, would you consider increasing your rating of our paper? If not, we would appreciate it if you could let us know what stands in the way of a stronger endorsement of acceptance on your part.

---

> > > ### Comment · Reviewer_EQPK · 2022-08-09
> > > **Response to the authors**
> > >
> > > Thank the authors for the response. My question on the connection between SAMPLR and OBL is addressed. However, I still think that the assumptions required by SAMPLR are quite strong. First, the assumption on the resettable simulator largely hides the exploration problem in RL. Second, the ground-truth distribution is often unknown to the learner and we can only draw finite samples from it. Besides, the experiments are insufficient to support the proposed algorithm. SAMPLR is only evaluated on two tasks. On one task, SAMPLR even reduces to naïve grounding. Therefore, I will retain my score.

---

> > > > ### Author Response · Authors · 2022-08-09
> > > > **We appreciate your feedback, but many of your comments are false**
> > > >
> > > > Thank you for letting us know your remaining concerns. We strongly disagree with these criticisms and feel it is important to push back on many of these claims which are false
> > > >
> > > > ### Resets remove the exploration problem
> > > > The statement that "the resettable simulator largely hides the exploration problem in RL" is incorrect. **Using resets in this way is NOT equivalent to injecting a priori knowledge about which regions of the state-space have high reward**, as your comment seems to imply. Automatic curriculum learning methods like SAMPLR, PLR, ACCEL, PAIRED, and ALP-GMM (which necessarily make use of resets), do not remove the need for exploration. On the contrary, these methods are the means by which we perform exploration over the space of environment configurations, seeking those environments that provide the most informative experiences for improving the current student policy—an especially challenging exploration problem as the student policy is non-stationary during training. Moreover, the student must still perform exploration across training environments to learn an optimal policy—training on a curriculum does not change this fact. Lastly, as we mentioned, with numerous references to well-regarded prior works, the assumption of a resettable simulator is fairly common when training in simulation. There is nothing inherently wrong with using a resettable simulator.
> > > >
> > > > ### The experiments do not support our method
> > > > We are also confused as to why you believe the experiments insufficiently support our method. It is true that in the Fruit Choice experiment, SAMPLR is equivalent to Naive Grounding, but the goal of this experiment was not to show the effectiveness of the full SAMPLR method. Rather, **this first experiment's purpose is to provide a simple initial example to show that curriculum-induced covariate shifts (CICS) can result in learning suboptimal policies, and that this problem can be solved through grounding the training distribution.**
> > > >
> > > > You then seem to completely ignore our main experiments in the challenging CarRacing with black ice environment, in which **we show that SAMPLR outperforms all baselines, including even PLR with naive grounding—thus demonstrating that naive grounding in itself is not sufficient to produce a robust policy in more complex domains** (Note that PLR is the state-of-the-art method for this CarRacing benchmark _without_ black ice). Further, we want to point out that this CarRacing environment requires solving a high-dimensional continuous-control task from pixels, and our specific experimental setting is harder than the versions of this environment in prior work: The aleatoric icy tiles make driving much more challenging, while our zero-shot transfer evaluation requires generalization to longer and more complex tracks replicating real Formula 1 tracks.
> > > >
> > > > ### Our method is not useful because we sometimes do not know the ground-truth distribution
> > > > As we clearly pointed out in our initial rebuttal, the ground-truth distribution is indeed known in many settings, and for this case, our method provides the first solution for using this information to ensure policies trained under a curriculum remain optimal under the ground-truth distribution. In the case of having to first learn the ground-truth distribution, **your criticism of the need to learn from a finite number of samples can be applied to any statistical method seeking to model a distribution based on real world data.** A principal aim of modern statistics is of course to model distributions of data based on finite samples. Any such method can be used to estimate the ground-truth distribution, which can then be used in SAMPLR.

---

> > > > > ### Comment · Reviewer_EQPK · 2022-08-10
> > > > > **Response to the authors**
> > > > >
> > > > > Thanks for the clarification. Most of my concerns are addressed. Thus, I will increase my score to 6. Below are some comments.
> > > > >
> > > > >
> > > > > I am sorry for the misleading claim that resets remove the exploration problem. My point is that the resettable simulator can largely help overcome the exploration issue. In particular, when the agent explores a new state, we can reset the agent to the new explored state many times to further expand the exploration region. In that case, the agent does not necessarily start from the initial state each time, which could improve the efficiency.

---

> ### Author Response · Authors · 2022-07-30
> **Response from authors, 1/2**
>
> Thank you for the positive comments about our work. We are glad you agree that curriculum-induced covariate shift (CICS), which we introduce in this paper, is an important problem to address. Further, it seems you agree our method successfully addresses this issue on two tasks, spanning both discrete and continuous control.
>
> Given your overall positive view of this work, we hope we can address your concerns around some of the assumptions and experimental choices made in this paper, and that you will consider increasing your rating of our work:
>
> ### Assumption of a resettable simulator
> We believe the assumption of a resettable simulator does not seriously limit the applicability of our method. In reinforcement learning, agents are typically trained in simulation, due to the costs (e.g. financial, temporal) and potential dangers of performing agent training in the wild. Since training largely occurs in simulation, and most simulators either support state-based resets or are easily modified to support such resets, SAMPLR can be widely applied. The additional work required to modify the simulator to support such resets is not unlike the additional work needed to deploy model-based RL on such environments. In both cases we are effectively extending the training environment with some additional logic. Moreover, these extensions are irrelevant at test-time, where we only care about the performance of the resulting policy.
>
> **Importantly, the problem setting of unsupervised environment design (UED) considered in our work assumes a controllable simulator, as it assumes that the environment configuration is controllable by the training process.** Thus the assumption of a resettable simulator is made by nearly all works in this area [1,2,3,4,5], in addition to many other impactful works [6,7], all of which were published in top-tier venues.
>
> ### References
>
> [1] Dennis, Michael, et al. "Emergent complexity and zero-shot transfer via unsupervised environment design." _Advances in neural information processing systems_ 33 (2020): 13049-13061.
>
> [2] Gur, Izzeddin, et al. "Environment generation for zero-shot compositional reinforcement learning." _Advances in Neural Information Processing Systems_ 34 (2021): 4157-4169.
>
> [3] Parker-Holder, Jack, et al. "Evolving Curricula with Regret-Based Environment Design." _International Conference on Machine Learning_. PMLR, 2022.
>
> [4] Pinto, Lerrel, et al. "Robust adversarial reinforcement learning." _International Conference on Machine Learning_. PMLR, 2017.
>
> [5] Portelas, Rémy, et al. "Teacher algorithms for curriculum learning of deep rl in continuously parameterized environments." _Conference on Robot Learning_. PMLR, 2020.
>
> [6] Ecoffet, Adrien, et al. "First return, then explore." _Nature_ 590.7847 (2021): 580-586.
>
> [7] Jiang, Minqi, Edward Grefenstette, and Tim Rocktäschel. "Prioritized level replay." _International Conference on Machine Learning_. PMLR, 2021.

---

### Official Review · Reviewer_dr8D · 2022-07-10

**Rating:** 5
**Confidence:** 3
**Soundness:** 2 fair
**Presentation:** 2 fair
**Contribution:** 2 fair

**Summary:**

This paper aims to generate curricula to 1) preserve the performance under the target distribution, and 2) improve the generalization to multiple levels by avoiding over-conservativeness due to the worst-case scenario. An algorithm is proposed based on Prioritized Level Replay and Bayesian inference. Some theoretical results are presented to give insights into the practical implementation. The experiment is conducted on a gridworld hard-exploration task and a continuous control task.

**Questions:**

See Weaknesses 2, 3 and 5.

Additional question:
How can SAMPLR be combined with UED methods other than PLR, such as PAIRED, RARL [1], etc.?

[1] Pinto, Lerrel, et al. "Robust adversarial reinforcement learning." International Conference on Machine Learning. PMLR, 2017

**Limitations:**

I see some of the limitations are discussed in the paper. I do not see any potential negative societal impact.

**Strengths And Weaknesses:**

**Strengths:**
This work looks at an interesting extension of Unsupervised Environment Design, which is associated with a target distribution of environments. The perspective is novel and intriguing. The experimental results seem to be significant.

**Weaknesses:**
1. The assumption to have an access to full access to the exact perfect simulator is very restrictive. It requires not only the parameter of the environment but the entirety of transition dynamics and initialization, random seed, and the ability to reset to any state. Most of the curriculum RL assumes access to the parameter of the environment, but not the latter parts.
2. Could the author provide more intuition about how the fictitious trajectories help mitigate the curriculum-induced covariate shift? I do not think the explanation is sufficient in the current draft.
3. Could learning a posterior $\bar{P}\left(\theta^{\prime} \mid \tau\right)$ using supervised learning require tons of data? The curriculum itself could produce limited pairs of trajectories and $\theta$. Are the data/environment interactions required to obtain this estimator included in the plots?
4. Although the objective is to generate a curriculum, there is no visualization of what kind of curriculum is generated and how they improve the performance. If $\theta$ controls the goal identity, showing the number of training rooms (Figure 3) is more of the result due to the curriculum but not how SAMPLR helps. Similar arguments go to Figure 4.
5. How is SAMPLR scaled to environments with a higher dimension of environment parameters? Both of the experiments have only one dimension and it is hard to see the potential.

---

> ### Author Response · Authors · 2022-07-30
> **Response from authors, 3/3**
>
> ### Scaling beyond one parameter
> We would like to point out that the environments in question are actually high-dimensional in terms of free parameters that can be varied by the curriculum. In Stochastic Fruit Choice, these parameters include the correct choice of fruit in addition to the size and arrangement of rooms, the fruit locations, and the start location of the agent. In CarRacing, there are approximately **1000 free parameters**, corresponding to whether each track tile is icy and the $12\times2=24$  parameters determining the location of the 12 control points of the Bézier curve defining the track.
>
> Regarding the belief state in particular, we reiterate our response to Reviewer tyft, who also inquired on this point: Modeling high-dimensional belief states can certainly pose challenges. However, **learning the belief model is an independent problem from the problem of correcting for CICS**. It is only this latter covariate shift problem that our paper seeks to define and solve. Thus, **the challenges of scaling belief model learning to higher dimensions is independent of the core contributions of this paper**, namely (1) formalizing the problem of covariate shifts in curriculum learning for RL and (2) proposing a solution to this problem based on optimizing Equation 1 using fictitious transitions. We believe CICS is an important type of distributional shift that has never before been formally characterized, making our work an important first step in both defining it and providing a first demonstration of how it can be successfully addressed. Thus we believe (1) and (2) form a valuable contribution to the RL literature. It seems you agree that CICS is an important and interesting problem for curriculum learning and environment design, and as such, we hope you would thus also advocate for the acceptance of this work based on these merits. Additionally, it is worth noting that scaling up high-dimensional belief models has been successfully accomplished in prior work on Off-Belief Learning [9], which like SAMPLR, trains using the same fictitious transitions based on a learned belief model in order to correct an analogous form of distributional shift in the cooperative multi-agent setting. OBL successfully scales fictitious transitions based on such a belief model to the multi-dimensional hidden state space of Hanabi, demonstrating that this general approach can scale up to multiple aleatoric parameters---more so, that the approach can even scale to modeling beliefs in complex multi-agent settings.
>
> ### How can SAMPLR be combined with other methods?
> The solution introduced in our paper for mitigating CICS replaces training transitions under the curriculum with transitions that are consistent with both the ground-truth distribution over the aleatoric parameters and the current trajectory. Just as SAMPLR implements PLR with training transitions replaced with such fictitious transitions, other curriculum methods can in principle incorporate the same fictitious transitions into their training loop. All that is required is to implement the same belief model as used by SAMPLR and replacing the training transitions with the fictitious transitions sampled under the belief model.
>
> Thank you also for pointing out RARL, which we now cite in the updated manuscript.
>
> Given these additional clarifications, which we are happy to incorporate into our camera-ready, would you consider increasing your support for our paper to "Accept"?
>
> ### References
>
> [1] Dennis, Michael, et al. "Emergent complexity and zero-shot transfer via unsupervised environment design." _Advances in neural information processing systems_ 33 (2020): 13049-13061.
>
> [2] Gur, Izzeddin, et al. "Environment generation for zero-shot compositional reinforcement learning." _Advances in Neural Information Processing Systems_ 34 (2021): 4157-4169.
>
> [3] Parker-Holder, Jack, et al. "Evolving Curricula with Regret-Based Environment Design." _International Conference on Machine Learning_. PMLR, 2022.
>
> [4] Pinto, Lerrel, et al. "Robust adversarial reinforcement learning." _International Conference on Machine Learning_. PMLR, 2017.
>
> [5] Portelas, Rémy, et al. "Teacher algorithms for curriculum learning of deep rl in continuously parameterized environments." _Conference on Robot Learning_. PMLR, 2020.
>
> [6] Jiang, Minqi, et al. "Replay-guided adversarial environment design."_Advances in Neural Information Processing Systems_34 (2021): 1884-1897.
>
> [7] Ecoffet, Adrien, et al. "First return, then explore." _Nature_ 590.7847 (2021): 580-586.
>
> [8] Jiang, Minqi, Edward Grefenstette, and Tim Rocktäschel. "Prioritized level replay." _International Conference on Machine Learning_. PMLR, 2021.
>
> [9] Hu, Hengyuan, et al. "Off-belief learning." _International Conference on Machine Learning_. PMLR, 2021.

---

> ### Author Response · Authors · 2022-07-30
> **Response from authors, 2/3**
>
> ### How do fictitious transitions mitigate CICS?
> Intuitively, the fictitious transitions mitigate CICS by forcing the student agent to optimize for reward transitions that are consistent with the ground-truth distribution over the aleatoric parameters---which otherwise can suffer covariate shifts under a curriculum. You can then think of the curriculum as bringing the agent to challenging states in the environment. This means the state distribution under the curriculum will differ from the state distribution experienced when sampling levels from the ground-truth distribution, and therefore the distribution over the experienced transitions will also differ. SAMPLR's fictitious transitions force the distribution of transitions at training to match that which would be experienced under the ground-truth distribution, if starting from the action-observation history at each time $t$ (i.e. the trajectory $\tau_t$). Thus, the student's learned policy will be optimal for the ground-truth distribution from each state, but the set of states actually experienced during training can be sampled from a biased curriculum distribution. This allows the curriculum to oversample more challenging states, e.g. track segments with more ice, in order to make the policy more robust to them, while still ensuring the policy is optimized for the ground-truth distribution, where ice appears rarely. If this intuitive explanation clarifies things for you, we would be happy to update the manuscript to include it in Section 4.
>
> ### Does learning the posterior $P(\theta | \tau)$ require tons of data?
> Note that SAMPLR does not seek to learn the full posterior over the environment configuration parameters, but only that over the subset of parameters that are aleatoric, i.e. $\Theta'$. You are correct that in general, learning $P(\theta'|\tau)$ can require many samples. However, this potential limitation—which is highly domain-dependent—is a limitation of model learning in general, rather than a specific limitation of our method. Developing algorithms for sample efficient posterior inference forms an important area of research in its own right, and these problems are independent of the problem of CICS that we define and seek to address in this work. Given that posterior inference is an orthogonal concern to the problem solved by SAMPLR, future improvements in performing such posterior inference can be substituted into SAMPLR, improving its performance. Moreover, training in simulation makes it cheap to train on a large number of samples, compared to training in the real world—otherwise the premise of environment design, which relies on collecting lots of data from a controllable simulator, would make little sense.
>
> In our experiments, the posterior is inferred online using the agent's trajectories during training (and thus does not use any additional data).
>
> ### Showing the curriculum over time
> We would like to push back on the statement that the "objective is to generate a curriculum." This is of course the goal of curriculum learning methods, but the aim of the fictitious transitions of SAMPLR is to correct for any harmful covariate shifts causing bias in existing curriculum learning methods, e.g. PLR. When uncorrected, these covariate shifts can cause curriculum learning to fall into degenerate outcomes, e.g. the teacher can just randomize between the two goals to maximally confuse the agent instead of challenging the agent by gradually increasing the number of rooms. As shown in Figure 3, in these settings standard UED methods (such as PLR) fail to produce a useful curriculum, while SAMPLR succeeds.
>
> The exact method by which SAMPLR fixes this issue is by ensuring the transitions experienced during training are consistent with the ground-truth distribution over aleatoric parameters, e.g. the correct goal. This prevents the teacher from being able to randomly switch the goal choice to maximize regret. Instead, the teacher must increase regret by adapting the actual difficulty of the levels, e.g. gradually increasing the number of rooms. In Figure 6 of Appendix B, We specifically show that with SAMPLR, the proportion of levels presented by the teacher where apples are the correct goal matches exactly the ground-truth distribution of these levels, for various settings of the true proportion of apple levels, i.e. $q = 0.7, 0.5, 0.3, showing that SAMPLR indeed fixes the aleatoric parameters to the ground-truth distribution. Similarly, in the CarRacing experiments, we see in Figure 4 that under SAMPLR, the curriculum over the average proportion of ice on training levels approaches the expected average under the ground-truth distribution, while the baselines consistently undersample ice---in this case, likely because it is easier for the adversarial teacher to induce higher regret on less icy tracks, where the student can more easily reach higher returns. As a result, policies learned under SAMPLR are more robust against higher levels of ice.

---

> ### Author Response · Authors · 2022-07-30
> **Response from authors, 1/3**
>
> Thank you for your review. We appreciate that you find our work to be an interesting extension of UED, that the perspective we provide is novel, and that our results are significant. Given your positive comments, we wonder if you would be willing to increase your rating to “Accept,” if we can address the concerns you raised.
>
> ### Assumption of a fully-controllable simulator
> In the CarRacing experiments we made a conscious design choice to use a simulator capable of state-based resets. This is a common assumption in many well-regarded works within curriculum learning [1,2,3,4,5,6] and other impactful works [7,8], all of which were published in top-tier venues. This assumption is commonly made and most often valid, because in practice, RL training largely occurs in simulation, e.g. in sim2real, sim2sim, and unsupervised environment design, which **requires such a controllable simulator as a core premise of the problem setting**.
>
> Using a simulator in this way allows us to accurately measure the effectiveness of our proposed solution for correcting curriculum-induced covariate shifts (CICS) independently of the effectiveness of *model learning*, i.e. learning the transition and reward functions which are required to perform accurate belief state modeling.
>
> The problem of learning the transition and reward functions is itself a challenging task, and scaling up methods for learning these functions to general POMDPs poses orthogonal research questions to those addressed by our paper. In particular, model learning can introduce errors in the learned transition and reward functions. Therefore, in order to assess the correctness of our proposed solution to CICS independently of the effectiveness of transition and reward function modeling, we chose to remove the impact of this latter problem in the CarRacing experiments by replacing the model with a parallel simulator that serves as a proxy for a perfect model of the transition and reward dynamics of the environment. Note that we still perform belief modeling by learning a posterior distribution over the belief state, assuming such a perfect transition and reward model. In general, SAMPLR does not require such a controllable simulator, as a learned model can be used instead.
>
> It is important to reiterate that, practically speaking, the problem setting of environment design directly requires a controllable simulator, as it requires that the environment configuration is controllable by the training process. When training inside a simulator (e.g. for environment design, sim2real, sim2sim), it makes sense to maximally exploit the simulator by performing state-based resets and accessing the true transition and reward functions. Most simulators are highly amenable to being modified to provide such privileged information and reset functionality during training. This is no different from performing such actions inside of a world model in model-based RL, with the only difference being that in this setting, the world model is a perfect model of the environment, i.e. the simulator. Using such privileged information and resets in simulation at training time is similar to providing a value function with privileged information when training actor-critic methods. In both cases, the model making use of the privileged information is removed at test-time. Such methods make maximal use of the available resources at training to produce more capable policies at test time.
>
> Of course, we do not always have access to such a simulator, and this specific limitation is mentioned in Section 4 (L203-207), in which case a learned model can be used instead. This constitutes a separate problem setting that can be considered for future extensions of our work.

---

> > ### Comment · Reviewer_dr8D · 2022-08-04
> > **Response to authors**
> >
> > Thanks for the reply. Here are my thoughts:
> >
> > > This is a common assumption in many well-regarded works within curriculum learning [1,2,3,4,5,6] and other impactful works [7,8]".
> >
> > As I stated in the original review, most of curriculum learning methods indeed require access to the simulator's parameters, such as RARL [4] (friction and mass), PAIRED [1] (forces and maze map), ALP-GMM [5] (shape of the obstacles). However, they DO NOT require state-based reset (the ability to reset to a specific state, e.g. joint configuration of the robot). I am not sure about other papers cited in the response, but feel free to point out which one has similar assumptions.

---

> > > ### Author Response · Authors · 2022-08-04
> > > **Regarding state-based resets**
> > >
> > > Thank you for clarifying your concern over our use of state-based resets. Some of the prior works we list do make explicit assumptions about state-based resets, or as we will argue, a set of equivalent assumptions (i.e. resets based on environment configuration and a fixed seed). Further, many popular RL environments directly support state-based resets (e.g. ALE, MuJoCo, DeepMind Control Suite).
> > >
> > > However, independent of the above facts, there is no reason NOT to use state-based resets when training RL in simulation, if exploiting such resets results in more robust policies. When we train in simulation, methods should ideally make maximal use of the simulator's affordances.
> > > Methods that exploit state-based resets, like SAMPLR, provide a means to get more value out of a simulator during training. This same argument in favor of state-based resets for training in simulation is made by the authors of Go-Explore, a breakthrough work in exploration for RL that appeared in Nature. Moreover, we make clear in our paper that our method's setting assumes training in simulation. Thus, it is not clear why our assumption of state-based resets should disqualify our method as a useful contribution in any way.
> > >
> > > Please see below for a more detailed discussion around these points:
> > >
> > > ### Support for resetting to environment configuration + fixed seed is equivalent to support for state-based resets
> > > There is typically no distinction between resetting to a specific simulator state and resetting to a specific environment configuration with a fixed random seed (setting the specific seed is supported by largely all RL environments that use random number generation, including OpenAI Procgen, MuJoCo with sensor noise, the Box2D OpenAI Gym environments like CarRacing and BipedalWalker. Otherwise, it is highly trivial to extend the environment interface to support this). The equivalence holds due to the following fact: Given a state $s_t$ reached by taking the actions in trajectory $\tau_t$ in the environment configuration $\theta$ for a specific random seed, you can always return to exactly $s_t$ by resetting the simulator to configuration $\theta$ with the same random seed and stepping the simulator through the same actions $a_0,...,a_{t-1}$ in $\tau_t$ that led to $s_t$.
> > >
> > > ### Prior works assume either reset to environment configuration + fixed seed or state-based resets
> > > Methods like PLR (ICML 2021), Robust PLR (NeurIPS 2021), and ACCEL (ICML 2022) assume resets based on environment configuration and fixed random seed, while the best-performing version of Go-Explore explicitly assumes state-based resets (Nature).
> > >
> > > ### State-based resets are explicitly supported in many popular environments
> > > Many popular RL environments even have helper methods that directly enable state-based resets, such as Atari Learning Environment (ALE), MuJoCo, and DeepMind Control Suite. In ALE, you can use the methods `clone_state` and `restore_state` for performing such state-based resets. Similarly, MuJoCo provides `get_state` and `set_state` methods. Likewise, DeepMind Control Suite offers `get_state`, `set_state`, and `physics.reset` to enable state-based resets.
> > >
> > > Lastly, we want to emphasize that our method does not necessarily entail using state-based resets. As we mentioned in our rebuttal, we made use of state-based resets to avoid the need for model-based RL in our CarRacing experiments. This allows us to cleanly assess the impact of training on fictitious transitions (the main mechanism behind SAMPLR) independently from the challenges of performing accurate model learning. Nevertheless, as the above points show, using state-based resets is a largely viable strategy whenever you are training in a simulator.

---

> > > > ### Comment · Reviewer_dr8D · 2022-08-09
> > > > **Response to Authors**
> > > >
> > > > Thanks for the reply. It addressed the majority of my concerns. I am willing to raise my rating to "Borderline Accept".

---

> > > > > ### Author Response · Authors · 2022-08-09
> > > > > **Thank you for increasing your support of our work**
> > > > >
> > > > > Thank you for considering our rebuttal and increasing your rating for our paper! We truly appreciate it.
> > > > >
> > > > > Given that, as you say, our rebuttal addresses the majority of your concerns, could you let us know what is standing in the way of a stronger endorsement of acceptance on your part?

---

### Official Review · Reviewer_tyft · 2022-07-11

**Rating:** 6
**Confidence:** 3
**Soundness:** 3 good
**Presentation:** 2 fair
**Contribution:** 3 good

**Summary:**

This paper proposes a method that addresses the covariant shift problem that can happen between the training parameter distribution induced by curriculum and test-time parameters in adaptive curriculum learning methods. Naive choice would be to sample parameters from ground-truth distribution but this would lose the benefit of curriculum learning, so the paper proposes a method that adjusts the policy learning to be optimal with ground-truth simulator while still sampling environment parameters with some curriculums.

**Questions:**

- Improving clarity of the paper would be nice (see above for details)
- Experiments in more complex setups instead of simple setup of fruit choice or single-parameter setup in racing env would be helpful
- Clarification on the agent design and discussion on how it would affect the analysis and the messages in the paper would be helpful


**Limitations:**

Limitation or potential negative impact are not discussed.

**Strengths And Weaknesses:**

**Strengths**
- This paper investigates an interesting problem of environment design with curriculum
- Experimental results show that the proposed method can outperform baselines potentially suffering from covariate shift

**Weaknesses**
- One of weaknesses in the paper is in the clarity especially Introduction, where it is difficult to understand what 'grounding' means in the title and also throughout the paper. It was also a bit difficult to parse the last sections in introduction without having a prior knowledge about the area; I could understand it after looking at the definitions from method section. Making it much more abstract in intro or more formal would be helpful for readability.
- In Figure 1, it would be nice to explain what U means and matching this during training would help, in caption.
- In Equation 3, It's not clear to me what does $\bar{P}(\tau_{t}^{O}|\theta,\tau_{t}^{A})$ mean and what would this being equal to 1 mean?
- It's curious to see how the method would work when considering more complex parameter configurations. Currently it's just a single parameter setup and it would be nice to show that the method could cover more complex setups.
- Is PPO agent based on recurrent architecture? The reason why I ask this question is because it is a bit questionable whether the ice parameter is indeed aleatoric -- if the agent has an information about the previous observations and actions where actions were not applied then agent might have more information about the road even without the visible pixel observation in roads, which makes it difficult to call icy parameter as aleatoric? It would be nice to clarify this point.

---
[2022-08-05 Update] Increased the score (5->6) per the discussion during the rebuttal phase

---

> ### Author Response · Authors · 2022-07-30
> **Response from authors, 2/2**
>
> ### Scaling up SAMPLR
> Regarding scaling SAMPLR beyond one parameter, we would like to point out that the environment design actually occurs over **approximately 1000 parameters**, because the choice of including ice on each individual tile is an individually controllable design parameter by the adversarial teacher, i.e. they are free parameters $\Theta$ of the UPOMDP. In addition, the adversarial teacher also chooses the location of the 12 control points determining the Bézier curve representation of the track. These control points introduce another $12\times2 = 24$ free parameters. We used a Bayesian model to model the uncertainty around this large design space as a single distributional parameter that determines the probability that any given tile has ice. This Bayesian model itself assumes two additional learned hyperparameters (i.e. the shape parameters of a Beta prior). Our experimental results show that parameterizing aleatoric uncertainty in terms of a high-level statistic in this way is sufficient for SAMPLR to resolve issues around CICS in such an environment design problem of hundreds of free parameters, and suggests that a similar strategy can be applicable in many other domains.
>
> In practice, modeling high-dimensional belief states can certainly pose challenges. However, learning **the belief model is an independent problem from the problem of correcting for CICS. It is only the latter covariate shift problem that our paper seeks to define and solve**. Learning effective belief models forms a separate, active subfield of research [1,2,3,4,5,6,7]. Thus, **the challenges of scaling belief model learning to higher dimensions is independent of the core contributions of this paper**, which are (1) formalizing the problem of covariate shifts in curriculum learning for RL and (2) proposing a solution to this problem based on optimizing Equation 1 using fictitious transitions. We believe CICS is an important type of distributional shift that has never before been formally characterized, making our work an important first step in both defining it and providing a first demonstration of how it can be successfully addressed. Thus we believe (1) and (2) form a valuable contribution to the RL literature. It seems you agree that CICS is an important and interesting problem for curriculum learning and environment design, and as such, we hope you would thus also advocate for the acceptance of this work based on these merits. Additionally, it is worth noting that scaling up high-dimensional belief models has been successfully accomplished in prior work, such as Off-Belief Learning [8], which like SAMPLR, trains using the same fictitious transitions based on a learned belief model in order to correct an analogous form of distributional shift in the cooperative multi-agent setting. OBL successfully scales fictitious transitions based on such a belief model to the multi-dimensional hidden state space of Hanabi, demonstrating that this general approach can scale up to multiple aleatoric parameters---in fact, the approach can even scale to modeling beliefs in complex multi-agent settings.
>
> We hope our response and the improvements to clarity in the paper address your concerns. Please let us know if you have any further questions. Given this additional discussion, we hope you will consider raising your support for our paper.
>
> References
>
> [1] Venkatraman, Arun, et al. "Predictive-state decoders: Encoding the future into recurrent networks." _Advances in Neural Information Processing Systems_ 30 (2017).
>
> [2] Igl, Maximilian, et al. "Deep variational reinforcement learning for POMDPs." _International Conference on Machine Learning_. PMLR, 2018.
>
> [3] Gregor, Karol, et al. "Temporal difference variational auto-encoder." _International Conference on Learning Representations_. 2019.
>
> [4] Sokota, Samuel, et al. "A Fine-Tuning Approach to Belief State Modeling." _International
> Conference on Learning Representations_. 2021.
>
> [5] Singh, Gautam, et al. "Structured world belief for reinforcement learning in pomdp." _International Conference on Machine Learning_. PMLR, 2021.
>
> [6] Hu, Hengyuan et al. "Learned Belief Search: Efficiently Improving Policies in Partially Observable Settings." AAAI, 2021.
>
> [7] Guo, Zhaohan Daniel, et al. "Neural predictive belief representations." _arXiv preprint arXiv:1811.06407_ (2018).
>
> [8] Hu, Hengyuan, et al. "Off-belief learning." _International Conference on Machine Learning_. PMLR, 2021.

---

> > ### Comment · Reviewer_tyft · 2022-08-05
> > **Response**
> >
> > Thanks for your response. Most of my concerns are addressed. I'm increasing the score to 6 (weak accept) with following comments:
> >
> > - Still curious to see what would happen if we would use the recurrent architecture to make the environment not partially observable in icy environment, and which environments & tasks with aleatoric parameters we could further consider.
> > - I understand that scaling up is a bit of independent problem and could be even feasible, but without supporting experiments, this still makes me think that a method is not practical yet to scale up to more complex setups with different parameter. This makes it difficult for me to recommend strong acceptance.

---

> > > ### Author Response · Authors · 2022-08-06
> > > **Much thanks for your increased support!**
> > >
> > > Many thanks for taking the time to consider our rebuttal and increasing your rating of our paper. We really appreciate it!
> > >
> > > We would like to point out that the decision-making problem is still partially observable with aleatoric uncertainty even with a recurrent policy. This is because the agent cannot know which *unvisited* track segments are icy. Thus, the locations of unvisited icy tiles remain aleatoric parameters at each point in the trajectory. In general, a capable adversarial teacher can then choose tracks that have little ice early on, but heavy ice later in the track (or vice versa) to mislead the agent and also thereby introduce a form of CICS.
> > >
> > > Regarding scaling up, we want to highlight that the separate results of Off-Belief Learning (OBL) in the multi-agent setting of Hanabi show that the fictitious transitions used by SAMPLR can be scaled up to higher dimensional belief states. OBL uses largely the same fictitious transition mechanism as SAMPLR to correct for a similar distributional shift in the cooperative multi-agent setting of Hanabi. We now include a detailed discussion relating SAMPLR to OBL in Appendix D.

---

> ### Author Response · Authors · 2022-07-30
> **Response from authors, 1/2**
>
> Thank you for your review. We are glad you agree that curriculum-induced covariate shifts (CICS), the new problem setting that we identify and formalize in this paper, is an “interesting problem”. Further, it seems you agree that our experiments show our proposed solution, SAMPLR, fixes this issue and outperforms baselines under such covariate shifts. We aim to address your open concerns in this rebuttal. Given your overall positive view of our work, we hope you will consider raising your rating to "Accept," if our response clarifies the points you raised in your review.
>
> ### Definition of grounding
> The exact meaning of *grounding the policy* used throughout the paper is defined in the Introduction on L56-58:
>
> > We can preserve optimality on $\overline{P}$ by grounding the policy---that is, ensuring that the agent acts optimally with respect to the ground-truth utility function for any action-observation history \tau and the implied ground-truth posterior over $\Theta$...[followed by Equation 1, which is what is optimized by the policy in order to perform this grounding].
>
> Further, we define the meaning of *grounding the training distribution* on L61-62:
>
> > We can ground the policy by grounding the training distribution, which means constraining the training distribution of aleatoric parameters $P(\Theta')$ to match $\overline{P}(\Theta')$.
>
> Throughout the paper, we use the term *ground truth* according to its standard definition, i.e. information that is known to be real or true.
>
> We are entirely open to rephrasing these definitions to be clearer, if you could let us know which parts in particular are confusing. In case the confusion is due to the definitions only appearing in the Introduction, our latest manuscript also reiterates these definitions in Section 3 for emphasis.
>
> ### Definition of $U$ in Figure 1
> As for the definition of U in Fig 1, this is the utility function defined in Equation 1, but with the expectation over $P$, so that $\theta \sim P(\theta|\tau)$, i.e. environment configurations (that is, levels) are sampled from the curriculum distribution $P$ rather than the ground-truth distribution $\overline{P}$. Recall that the important difference between $P$ and $\overline{P}$ is that $P$ is the curriculum distribution of aleatoric parameters $\Theta'$ (e.g. whether apple or banana is the correct choice), and so under $P$, the aleatoric parameters can suffer from a covariate shift with respect to their ground truth distribution. In contrast, $\overline{P}$ is the ground truth distribution, so under $\overline{P}$, the aleatoric parameters are fixed to their ground-truth distribution. Thus, the diagram simply shows that without applying the correction via fictitious transitions used by SAMPLR, a curriculum can cause covariate shifts that lead the agent to optimize for a utility function $U$ that is not based on the ground-truth distribution $\overline{P}$, and therefore the resulting policy can be suboptimal under $\overline{P}$. Applying SAMPLR corrects for this bias, leading the agent to optimize for the ground-truth utility function, i.e. $\overline{U}$, which we show preserves optimal performance on the ground-truth distribution $\overline{P}$. We have updated the caption in Fig 1 to make the definition and distinction between $U$ and $\overline{U}$ clear.
>
> ### Notation in Equation 3
> Thank you for bringing up this notational issue. We agree the $P(\tau_t^O | \theta, \tau_t^A)$ notation was confusing. Our updated manuscript removes this notation altogether. Specifically  $P(\tau_t^O | \theta, \tau_t^A)$ has been replaced by $P(\tau_t | \theta)$, which is the probability of the current action-observation history $\tau_t$ given a specific environment configuration $\theta$, i.e. an instantiation of the UPDOMP's free parameters.
>
> ### Agent architecture
> We would like to confirm that the agent policy itself is not recurrent, but rather uses an observation based on stacking only the last 4 frames. The full details of the agent architecture, as well as choice of hyperparameters and other training details, are provided in Appendix C. In particular, details for the CarRacing experiments are in Appendix C.2.
>
> ### Limitations and potential negative impact
> We would also like to point out that we do explicitly state the main limitations of our method in Section 4 (L203-207). A broader impact statement is also included in Appendix E.

---

### Author Response · Authors · 2022-08-08
**Reviewers, let us know if we have addressed your concerns (especially Reviewer dr8D and Reviewer EQPK)**

Many thanks to all reviewers for taking the time to review our paper and share useful feedback. As the author-reviewer discussion phase comes to a close tomorrow, we are wondering if you could please let us know if our rebuttal addresses your remaining questions and concerns, and whether you would consider increasing your rating of our paper in light of our responses. If not, we would really appreciate if you could share what more stands in the way of your actively supporting our paper for acceptance.

In particular, thank you, **Reviewer tyft**, for engaging with our rebuttal during the discussion phase. We appreciate your increasing your rating of our paper. We provided further details in our response to your latest comment that should address your follow-up comments and instill further confidence in our work.

We hope that **Reviewer dr8D** and **Reviewer EQPK** will similarly engage with our rebuttal before the author-reviewer discussion phase ends tomorrow. In particular, our responses directly addressed their main concerns, namely the following:

- **The assumption of a resettable simulator:** We reiterate the fact that the problem setting of unsupervised environment design, considered in this work, directly assumes (in fact, requires) a resettable simulator. Further, regarding state-based resets: There is nothing inherently wrong about making use of them when they are available. When training in simulation, state-based resets combined with methods like SAMPLR provide a means to produce more robust final policies at test time—which is the goal of this work, like that of related, preceding works (e.g. PAIRED, PLR, ACCEL). Moreover, state-based resets are directly supported by many popular RL environments, including Atari Learning Environment, MuJoCo, and DeepMind Control Suite. Our rebuttal also provided many examples of prior works published in top-tier venues making this same assumption of a resettable simulator capable of state-based resets, thus showing we are not alone in making this valid (and in fact, somewhat common) assumption, when training in simulation.

- **Scaling to higher-dimensional settings:** As discussed in our responses, the design problems considered in our experiments are indeed high dimensional. In the CarRacing experiments, the adversarial teacher must choose values for approximately 1000 free parameters per level (i.e. for each track), because whether each individual track tile contains ice is a free parameter of the underspecified POMDP. Similarly, the fruit choice levels require designing over a high-dimensional design space encompassing the number of rooms, their individual sizes, their arrangement, the start location of the agent, the location of each piece of fruit, and the identity of the correct choice of fruit. Furthermore, we emphasize that **the problem of scaling the belief model to high-dimensional belief states is an entirely separate problem from the core aims of this paper**, which are (1) formalizing the problem of covariate shifts in curriculum learning for RL and (2) proposing a solution to this problem based on optimizing Equation 1 using fictitious transitions. Our work accomplishes both of these aims. Curriculum-induced covariate shift (CICS) is an important type of distributional shift that has never before been formally characterized, making our work an important first step in both defining it and providing a first demonstration of how it can be successfully addressed. Thus we believe (1) and (2) form a valuable contribution to the RL literature, and we hope the reviewers will consider accepting our work based on these merits.

The above points provide a brief summary of the full discussion in our responses. We took much care and effort in deeply considering these reviews and writing our responses to address each concern raised by the reviewers. We hope that **Reviewer dr8D** and **Reviewer EQPK** will find the time to similarly consider our responses to the questions and comments that they raised.

---

### Meta-Review · Area_Chair_hoRm · 2022-08-25

**Recommendation:** Accept
**Confidence:** Less certain

**Metareview:**

This work proposes a prioritized level replay and Bayesian inference based algorithm for better generation of curricula via unsupervised environment design. It tries to address the problem of covariate shift induced by curriculum itself with respect to the test distribution. Overall this has been well-received by reviewers. There was rich discussion about whether the assumption of a resettable controller is overly restrictive. The authors have convincingly responded that not only is it necessary but should be taken advantage of wherever available and that many popular RL environments provide reset capability to desired states. The gist of this discussion would do well to find an explicit place in the discussion section of the camera-ready version of the paper.

**Award:**

No

---

### Decision · Program_Chairs · 2022-09-14

Accept